



# The utility of simulated ocean chlorophyll observations: a case study with the Chlorophyll Observation Simulator Package (version 1) in CESMv2.2

Genevieve L. Clow[1,2,3], Nicole S. Lovenduski[1,2], Michael N. Levy[4], Keith Lindsay[4], and Jennifer E. Kay[1,3]

[1]Department of Atmospheric and Oceanic Sciences, University of Colorado Boulder, Boulder, Colorado, USA
[2]Institute of Arctic and Alpine Research, University of Colorado Boulder, Boulder, Colorado, USA
[3]Cooperative Institute for Research in Environmental Sciences, University of Colorado Boulder, Boulder, Colorado, USA
[4]Climate and Global Dynamics Laboratory, National Center for Atmospheric Research, Boulder, Colorado, USA

**Correspondence:** Genevieve Clow (genevieve.clow@colorado.edu)

**Abstract.** For several decades, a suite of satellite sensors has enabled us to study the global spatiotemporal distribution of phytoplankton through remote sensing of chlorophyll. However, the satellite record has extensive missing data, partially due to cloud cover; regions characterized by the highest phytoplankton abundance are also some of the cloudiest. To quantify potential sampling biases due to missing data, we developed a satellite simulator for ocean chlorophyll in the Community Earth System

Model (CESM) that mimics what a satellite would detect if it were present in the model-generated world. Our Chlorophyll Observation Simulator Package (ChlOSP) generates synthetic chlorophyll observations at model runtime. ChlOSP accounts for missing data – due to low light, sea ice, and cloud cover – and it can implement swath sampling. Results from a 50-year pre-industrial control simulation of CESM-ChlOSP suggest that missing data impacts the apparent mean state and variability of chlorophyll. The simulated observations exhibit a nearly -20 % difference in global mean chlorophyll compared with the

standard model output, which is the same order of magnitude as the projected change in chlorophyll by the end of the century. Additionally, missing data impacts the apparent seasonal cycle of chlorophyll in subpolar regions. We highlight four potential future applications of ChlOSP: (1) refined model tuning, (2) evaluating chlorophyll-based NPP algorithms, (3) revised time to emergence of anthropogenic chlorophyll trends, and (4) a testbed for the assessment of gap-filling approaches for missing satellite chlorophyll data.

## 1   Introduction

The spatiotemporal distribution of marine phytoplankton, unicellular algae responsible for ∼50 % of global net primary production, greatly impacts fisheries, ecosystems, and the marine carbon cycle (Chassot et al., 2010; Fay and McKinley, 2017). Phytoplankton growth is dependent on temperature, light, and nutrient availability. Regions characterized by upwelling of nutrient-rich water, such as the equatorial, subpolar, and eastern boundary current regions, are some the most biologically

productive (Siegel et al., 2013). In subpolar regions, wintertime mixing brings nutrients to the surface, but the lack of sunlight prohibits growth until the spring. This results in a pronounced seasonal cycle in phytoplankton abundance at high latitudes. In





contrast, subtropical regions have abundant light but lack nutrients due to density stratification, which reduces vertical mixing. This nutrient limitation is relieved at the equator due to ocean dynamics, leading to elevated productivity throughout the year.

For over 20 years, a suite of satellite sensors has enabled us to study the global spatiotemporal distribution of phytoplank-
ton through remote sensing of chlorophyll-a (McClain, 2009; Siegel et al., 2013). Chlorophyll-a (hereafter, referred to as chlorophyll) is the primary photosynthetic molecule in plant cells, and it affects ocean spectral properties in identifiable wave-lengths that can be remotely observed by passive satellite spectroradiometers. Since remote sensing of chlorophyll relies on visible light, detection is not possible at night or beneath cloud cover and sea ice. The most commonly used algorithms to derive chlorophyll concentration from remote sensing reflectance relies on the ratio of blue to green wavelengths. The Hu and O'Reilly chlorophyll algorithms are based on empirical relationships between remote sensing reflectance and in situ measure-ments (O'Reilly et al., 1998; O'Reilly and Werdell, 2019; Hu et al., 2012). These measurements have provided a global dataset with which we can study phytoplankton abundance and variability.

Earth system models (ESMs) can generate projections of future phytoplankton abundance and productivity in a changing climate, and are thus a key tool for quantifying the impacts of changing climate on the carbon cycle (Wilson et al., 2022) and fisheries productivity (Tittensor et al., 2018), as well as the avoided impacts under climate change mitigation (Krumhardt et al., 2017). ESMs produce century-scale projections of climate using mathematical equations to describe atmospheric and oceanic processes, including a full terrestrial and ocean carbon cycle. ESMs simulate nutrient cycling in the ocean by accounting for the role of phytoplankton and their zooplankton predators. In simulating the abundance of phytoplankton, models include processes such as photosynthesis, respiration, grazing, and sinking. These biological terms depend on physical and chemical oceanography simulated by the model. ESM projections suggest that phytoplankton abundance is affected by anthropogenic climate change due to changes in stratification and the consequent nutrient and light availability brought on by upper ocean warming (Kwiatkowski et al., 2020). As the climate changes, oligotrophic regions are expected to see a decline in phytoplank-ton abundance, while regions with light-limited production are likely to see increases in abundance (Kwiatkowski et al., 2020; Marinov et al., 2010). These regional changes in primary productivity have critical implications for the coupled carbon-climate system, as well as for marine ecosystems and fisheries.

In order to use ESMs to project the future, we first need to validate them using present day observations. During the model development phase, the biogeochemistry component of ESMs are often tuned to the satellite record of chlorophyll. Based on simulated phytoplankton biomass, ESMs calculate the chlorophyll concentration at each time step and grid cell. The process of tuning involves making minor adjustments to various parameters so that the model outputs more closely align with observations. Some examples of ocean biogeochemical parameters are nutrient uptake rates, maximum grazing rates and growth efficiency coefficients for each phytoplankton functional group (Long et al., 2021). These parameter values are based on information provided from laboratory studies, but the exact values are not known. A typical tuning target involves minimizing error in the broad global patterns of the modeled climatology (Danabasoglu et al., 2020).

Due to the availability of surface chlorophyll data from satellite measurements, it is a convenient tuning target. However, ESM-produced chlorophyll is not identical to that estimated via satellite. ESMs provide a complete record of chlorophyll across the global ocean, whereas there are significant data gaps in the satellite record. Gregg and Casey (2007) estimated that

that satellite sampling bias leads to an 8 % overestimate of global mean chlorophyll. Additionally, ESMs calculate chlorophyll directly, while satellite estimates of chlorophyll are derived from optical properties of ocean water, leading to further uncertainty. As such, we may be tuning our ESMs to biased observations, which could inflate inaccuracies in ESM projections.
Previous validation efforts have tried to make models more satellite-like by generating remote sensing reflectances within the model (Dutkiewicz et al., 2018). However, one of the biggest causes of potential bias and model mismatch has not yet been addressed: the role of missing data.

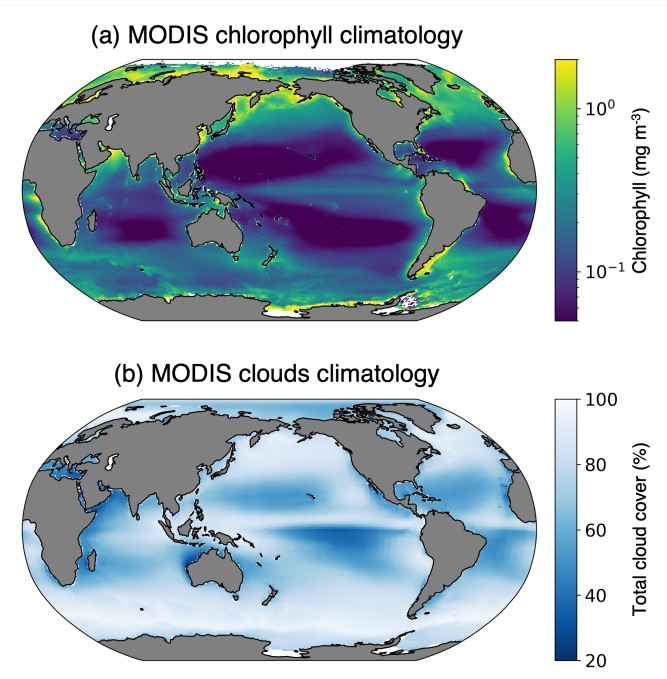

**Figure 1.** Multi-year means: (a) Aqua MODIS chlorophyll concentration entire mission composite image (2002–2023), (b) Aqua/Terra MODIS total cloud cover 2000–2011 mean.

Globally, the largest impediment to satellite chlorophyll detection arises from solar zenith angle limits (Gregg and Casey, 2007), which prevent detection at nighttime. This is especially important in the high latitude regions in wintertime, when sea ice
(a further detection challenge) is present. Another significant barrier to chlorophyll retrieval via satellite is cloud cover (Gregg and Casey, 2007; Mikelsons and Wang, 2019). On an average day, Moderate Resolution Imaging Spectroradiometer (MODIS) sensors aboard the Terra and Aqua satellites are unable to detect chlorophyll in approximately 72 % of the ocean's surface due to clouds (King et al., 2013). Some of the cloudiest ocean regions, such as the subpolar North Atlantic, North Pacific, and Southern Ocean, also have some of the highest rates of primary productivity (Fig. 1). The co-location of high chlorophyll and
cloud coverage results from atmospheric and oceanic dynamics: global wind patterns control the climatological distribution of clouds and ocean upwelling. Therefore, we are unable to reliably detect phytoplankton in the regions where they are most abundant.





To address the role of missing data in the detection of chlorophyll, we developed a satellite observation system simulator for ocean chlorophyll in the Community Earth System Model (CESM): the Chlorophyll Observation Simulator Package, ChlOSP

(Fig. 2). Using ChlOSP, CESM simultaneously generates an estimate of full-field chlorophyll and synthetic observations (obscured by simulated solar zenith angle, sea ice and clouds). This enables us to (1) estimate sampling biases due to cloud cover and other sources of missing data, and (2) make a more direct comparison between the model outputs and real-world data to improve model tuning. Here, we present initial results from a 50-year pre-industrial simulation using the simulator and briefly explore future applications of this new tool. As we will show, clouds can alter the apparent mean state, seasonality, and vari-

ability of chlorophyll. In addition to improving model tuning exercises, applications of ChlOSP include estimating the time of emergence of anthropogenic trends in the chlorophyll record, evaluating methods for calculating net primary productivity from satellite-observed chlorophyll, and creating a self-consistent gap-filling testbed.

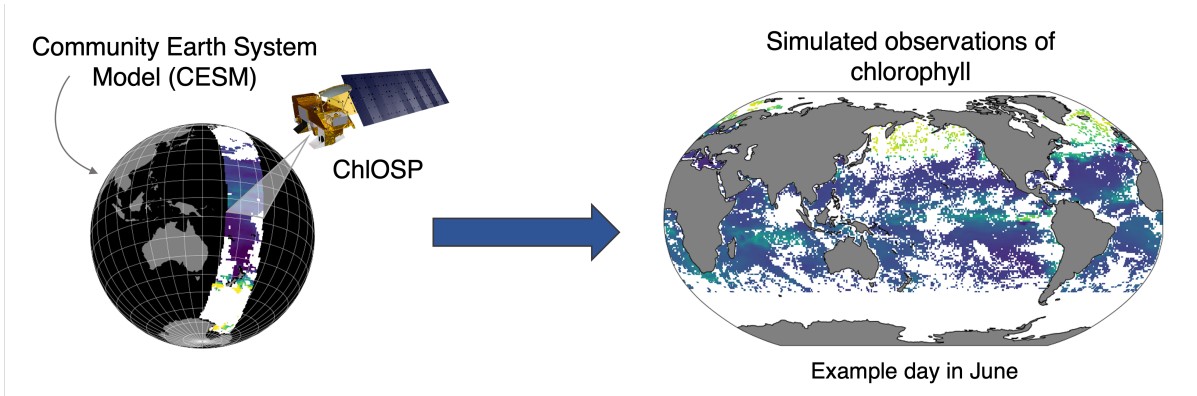

**Figure 2.** Conceptual diagram of ChlOSP.

## 2   Methods

### 2.1   Community Earth System Model version 2

The Community Earth System Model version 2 (hereafter, referred to as CESM) is a fully coupled global climate model developed at the National Center for Atmospheric Research (NCAR) (Danabasoglu et al., 2020). The model includes components for the ocean (POP2), atmosphere (CAM6), sea ice (CICE5), land (CLM5), land ice (CISM2), waves (WW3) and rivers (MOSART), which exchange information through the coupler (CPL7). The carbon cycle is represented through land and ocean biogeochemistry sub-components, which exchange carbon fluxes through the atmosphere. Here, we use version 2.2 of CESM,

which was tuned via parameter adjustment and expert evaluation to correct ocean biogeochemical biases (Yeager et al., 2022).





### 2.1.1 Ocean model

The ocean component in CESM is the Parallel Ocean Program Version 2 (POP2) (Danabasoglu et al., 2020; Smith et al., 2010). The coupler passes states and fluxes between CAM6 and POP2 at 30 minute and hourly intervals, respectively. The standard grid for POP2 has an approximately 1 degree horizontal resolution with 60 vertical levels ranging in thickness from 10 m at the surface to 250 m at depth. The model includes parameterizations of sub-grid scale processes, which are important for modeling ocean biogeochemistry. For example, bigeochemical tracers are impacted by parameterizations for eddy diffusivity, along with estuary, wave-driven and vertical mixing (Danabasoglu et al., 2020). CESM also includes sub-grid scale light availability, which impacts photosynthesis rates and improves the representation of phytoplankton in regions with sea ice (Long et al., 2015).

Biogeochemical ocean processes are modelled in CESM by the Marine Biogeochemistry Library (MARBL) (Long et al., 2021). We use a configuration of MARBL that simulates three phytoplankton functional groups: diatoms, diazotrophs, and small (pico/nano) phytoplankton. The growth term depends on light, nutrients, and temperature. The loss terms include aggregation/sinking and grazing, which is controlled by one zooplankton functional group. For each phytoplankton type, nutrient limitation is calculated based on phosphorus, nitrogen, iron and silicon tracers. Nutrient concentrations evolve through biological processes as well as nutrient fluxes from dust deposition and river inputs. The light limitation term is a function of photosynthetically active radiation (PAR), which is calculated as 45 % of incoming shortwave radiation at the surface. PAR varies with time of day, depth in the water column, cloud coverage, and sea ice. MARBL includes a dynamic chlorophyll:carbon ratio ($\theta$) within each phytoplankton group. The optimal $\theta$ depends on temperature, light and nutrient availability, allowing phytoplankton to adapt to their environment through the process of photo-acclimation (Geider et al., 1998). Since satellite observations of chlorophyll are used as a proxy for phytoplankton biomass, including the photo-acclimation term is key for estimating biases in the satellite record.

### 2.1.2 Atmosphere model

The atmosphere model used in CESM is the Community Atmosphere Model Version 6 (CAM6) (Danabasoglu et al., 2020). The default configuration for CAM6 is a finite volume dynamical core with a horizontal resolution of 1.25° in longitude and 0.9° in latitude. It has 32 vertical levels up to about 40 km in height. Clouds are simulated through parameterizations of the planetary boundary layer and shallow convection following the Cloud Layers Unified By Binormals method (Golaz et al., 2002; Bogenschutz et al., 2013). This allows for sub-grid scale variations in temperature, humidity and vertical velocity, leading to partial cloud cover within a grid cell. The cloud microphysics scheme is based on Gettelman and Morrison (2015), with ice nucleation depending on both temperature and aerosol concentration (Wang et al., 2014).

This work builds off existing satellite simulator software designed for clouds – the Cloud Feedback Model Intercomparison Project (CFMIP) Observation Simulator Package (COSP) (Bodas-Salcedo et al., 2011; Webb et al., 2017). Within CAM6, COSP provides model outputs that are directly comparable to real-world satellite observations (Pincus et al., 2012). This software package has been incorporated into many climate models (Klein et al., 2013) including CESM (Kay et al., 2012). The





latest version of COSP, COSP2, is functional in CESM version 2 (Swales et al., 2018). COSP simulates the observations

of several satellite sensors, including Multi-angle Imaging Spectroradiometer (MISR), Moderate Resolution Imaging Spectroradiometer (MODIS), CloudSat, Cloud-Aerosol Lidar and Infrared Pathfinder Satellite Observations (CALIPSO), and the International Satellite Cloud Climatology Project (ISCCP). In COSP, the atmospheric model grid cells are divided into internally homogeneous subcolumns that roughly correspond to the spatial resolution of satellite data, then forward modeling is applied to each subcolumn to generate satellite-like measurements. The results from each subcolumn are then aggregated

back to the model resolution. We developed ChlOSP using both the ISCCP and MODIS cloud simulators. Cloud properties are calculated at hourly radiation time step intervals using 250 subcolumns.

## 2.2 ChlOSP description

The goal of ChlOSP is to generate model output that is comparable to the NASA Ocean Color Level 3 chlorophyll concentration data product. Level 3 data is an imperfect estimate of the actual surface chlorophyll concentration due to atmospheric correction,

instrumentation, and ocean color algorithm uncertainties. Here, we focus on sampling biases that arise due to missing data by assuming that the satellite can detect the true ocean surface chlorophyll with 100 % accuracy in clear-sky conditions. We discuss the implications of this assumption in the Discussion and Conclusions section below.

Simulated observations of surface chlorophyll are generated within the ocean model component, POP2. ChlOSP calculates the sum of the surface (0–10 m) chlorophyll concentrations over all phytoplankton functional types. At each model time step,

POP2 uses multiple variables to calculate the chlorophyll weights, which represent the fraction of each model grid cell that would be viewable by a satellite. The weighted chlorophyll field can then be compared with the total surface chlorophyll to assess biases due to missing data.

### 2.2.1 Calculation of the weights

ChlOSP accounts for clouds, sea ice, and low sunlight (high solar zenith angle), all of which prevent satellite detection. In the

default CESM configuration, sea ice fraction is calculated in CICE5 and then passed to POP2. For clouds and solar zenith angle, the CESM coupler is modified to pass these additional variables from CAM6 into POP2. Specifically, we use COSP-generated MODIS and ISCCP total cloud cover and cosine of the solar zenith angle. Since the quality of chlorophyll retrievals starts to decline at a solar zenith angle of about 70 degrees (Mikelsons and Wang, 2019), we apply a masking threshold of 0.342 for the cosine of the solar zenith angle.

Satellites have a high spatial resolution (4.6 km for MODIS) compared to the coarse model grid (1 degree). To account for the discrepancy in resolution, we apply a weighting method for sea ice and cloud cover. The weights range from 0 to 1, where 1 indicates that 100 % of the cell was viewable by the satellite and 0 indicates that no detection was possible. To calculate the weight from modeled sea ice and cloud cover fields, which are both expressed in terms of the fraction of a grid cell that is covered, these values are subtracted from 1. All weights are assumed to be independent from one another, so the final weight is

the product of the weights calculated from each input parameter. At every model time step, the surface chlorophyll is multiplied by the weights. Then, the weighted chlorophyll and the weights are both output by the model at the desired frequency.





### 2.2.2 Simulator configurations

In order to test the sensitivity of ChlOSP to the modeled representation of cloudiness, we generate outputs using two different simulated cloud observations. We also test the impact of sampling frequency by comparing results from sampling chloro-
phyll once-per-day vs. all sunlit time steps. Here, we present the results from three different configurations of ChlOSP: (1) all-daylight sampling with simulated ISCCP cloud observations, (2) all-daylight sampling with simulated MODIS cloud observations, and (3) 1:30 pm sampling with simulated MODIS cloud observations. The ISCCP configuration is comparable to a chlorophyll observing system that combines many satellites, while the 1:30 pm sampling of MODIS is more similar to observations from an individual satellite.

MODIS and ISCCP cloud cover are simulated observation fields generated in COSP. The simulated observations are generated using the same model information, but different cloud-detection algorithms result in different observed total cloud fraction (Bodas-Salcedo et al., 2011; Pincus et al., 2012). In the real world, the ISCCP cloud cover product combines data from multiple passive sensors, including geostationary weather satellites (Rossow and Schiffer, 1991), and the MODIS instrument is aboard two polar-orbiting satellites (Aqua and Terra). COSP in CESM samples each location at all time steps and does not include
varying orbits for the different satellite simulators. However, since both MODIS and ISCCP rely on visible wavelengths, only sunlit time steps are included (Kay et al., 2012). For the all-daylight ISCCP and MODIS configurations in ChlOSP, we similarly sample chlorophyll at all sunlit locations at each model time step.

    The two MODIS configurations can be compared to assess the impact of satellite-like sampling vs. sampling at all daylight time steps. Since phytoplankton and chlorophyll concentrations exhibit a diurnal cycle, the time of detection may impact the
results (Salisbury et al., 2021; O'Malley et al., 2014). We simulate a simplified version of NASA's Aqua orbit. Aqua is a polar-orbiting satellite that collects data at approximately 1:30 pm local time, with a swath width of 2300 km. On a given day, Aqua samples the poles several times but has data gaps at low latitudes because successive orbits are not aligned longitudinally. These low latitude gaps are then filled during an orbit on the subsequent day, and the orbital pattern repeats every 16 days. To simplify the complex orbital geometry, the simulated Aqua satellite in ChlOSP flies exactly the same orbit every day. The swaths are
vertically centered on local 1:30 pm and have a swath width of 1668 km. Since successive orbits are aligned, there are no inter-orbit gaps near the equator. While simplified, this method simulates the general sampling pattern of Aqua: approximately once per day at low latitudes with increasing frequency at higher latitudes (Fig. S1).

### 2.3 Initial simulation and analysis

### 2.3.1 CESM simulation setup

We tested ChlOSP in a pre-industrial control simulation. In this type of simulation, forcing fields (greenhouse gases, aerosols, etc.) are fixed at values for the year 1850. Therefore, fluctuations in the system are a result of internal climate variability, rather than a response to external forcing. After initializing the model, equilibration of the deep ocean can take hundreds of years. However, we are interested primarily in surface ocean variables, which reach equilibrium relatively quickly. In our model





simulation, equilibrium of surface chlorophyll is reached after approximately 15 years (Fig. S2). We ran the model for 50 years,
but only analyzed the last 30 years of data, which is the number of years commonly used to calculate chlorophyll climatologies.

### 2.3.2    Model outputs

Each configuration of ChlOSP generates a chlorophyll output along with the corresponding weights (see Table A1 for a complete list of new model outputs). ChlOSP outputs were added to a new hourly POP2 output file stream. We use these hourly model outputs to assess the impact of photo-acclimation and the diurnal cycle of chlorophyll. We expected that the diurnal
cycle may impact our MODIS 1:30 pm swath results, since this version only samples once per day.

In post-processing, the weights are used along with their corresponding variable to calculate means over space and time (see Appendix B for equations). To investigate sampling biases in synthetic observations, we calculate the chlorophyll climatology using three ChlOSP outputs: Standard, Clear-Sky, and Cloudy. The Standard climatology is the unweighted, standard model output (i.e. total surface chlorophyll). The Cloudy output includes daylight, sea ice, and cloud cover weights, and the Clear-
Sky version includes only the daylight and sea ice weights – allowing us to isolate the impact of cloud cover. The Cloudy and Clear-Sky outputs vary depending on which configuration of ChlOSP is used (i.e., ISCCP, MODIS, MODIS Swath).

When calculating the global mean of chlorophyll, we weight each grid cell by how frequently it was sampled (Equation B3). To do this, we apply the time-averaged weights, which represent the sample size for each grid point. Figure S3 shows the chlorophyll climatologies along with the corresponding time-mean of the weights for each cloudy configuration. Here, the
weights are the product of the simulator weights and the total area of each grid cell. The normalized weights represent the mean area seen by the satellite relative to other points on the globe.

### 2.3.3    Regions of interest

For our evaluation of ChlOSP and subsequent analysis, we focus on highly biologically productive and cloudy open-ocean regions, particularly those with large seasonal cycles. We use ocean biomes defined by Fay and McKinley (2014) (Fig. S4).
These regions were defined using observations of chlorophyll, sea surface temperature, ice fraction and mixed layer depth. The biomes that we highlight are the North Pacific (biome 2: North Pacific Subpolar Seasonally Stratified), North Atlantic (biome 9: North Atlantic Subpolar Seasonally Stratified), Arctic (biome 1: North Pacific Ice and biome 8: North Atlantic Ice), and Southern Ocean (biome 16: Southern Ocean Subpolar Seasonally Stratified). Note that the Arctic biome used in our analysis includes only regions that are seasonally ice-free and does not correspond to the entire Arctic ocean.

## 2.4    ChlOSP evaluation

Before using ChlOSP to quantify the impact of missing data on chlorophyll, we demonstrate that ChlOSP is able to realistically simulate observations. Here, we focus on evaluating how well the simulator mimics real-world satellite data by calculating the percentage of missing chlorophyll data during the sunlit period of the day. This metric was selected for evaluation as it



captures the efficacy of the simulator in an imperfect representation of the Earth system; CESM exhibits known biases in both

chlorophyll (Long et al., 2021) and clouds (Danabasoglu et al., 2020).

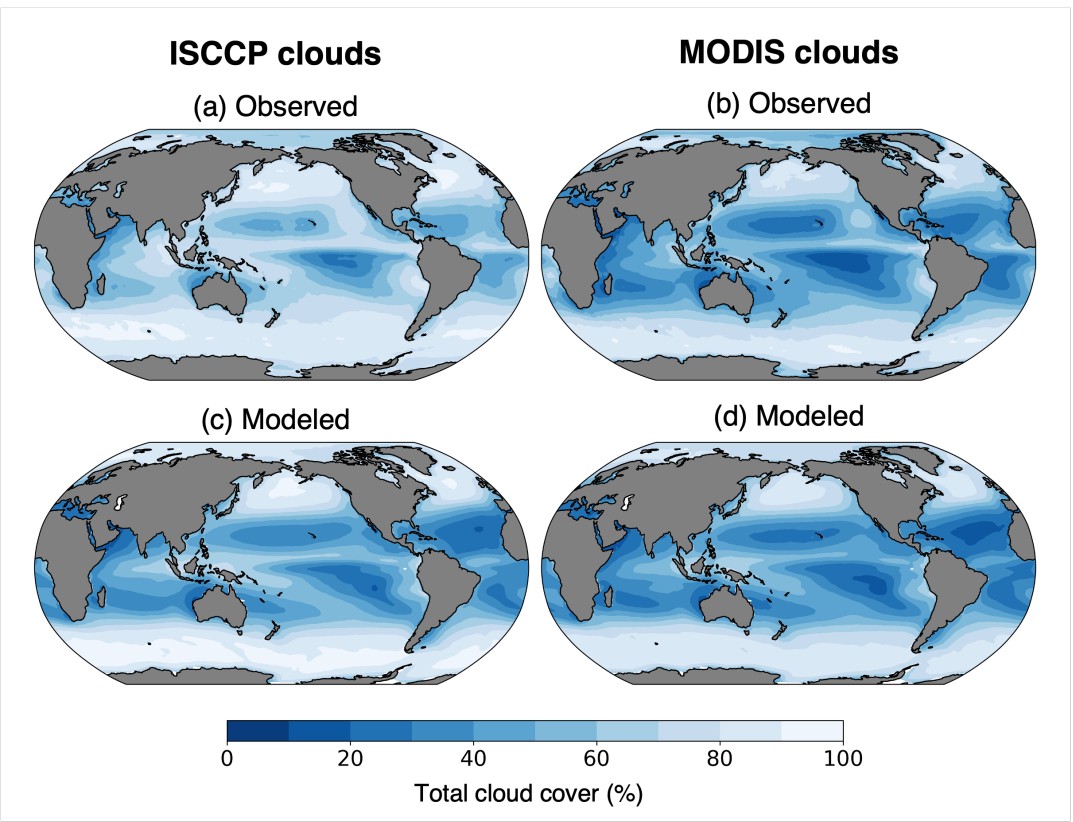

**Figure 3.** Cloud fraction climatology from CESM and observations. (a) ISCCP cloud cover observations from 2002 to 2016, (b) MODIS cloud cover (percent of pixels within 1 degree cell that had successful cloud property retrievals) from 2002 to 2016, (c) ISCCP and (d) MODIS modeled clouds from 30-year CESM pre-industrial simulation.)

Total cloud coverage greatly impacts the amount of missing chlorophyll data. Due to differing cloud detection algorithms, ISCCP and MODIS produce different estimates of total cloud cover (Fig. 3). The global mean cloud coverage from the ISCCP observations is approximately 13 % higher than the MODIS observations. The primary difference is in the treatment of partially cloudy pixels, which are treated as fully cloudy in the ISCCP simulator and fully clear in the MODIS simulator (Pincus et al.,

2012). Since partial cloud cover within a pixel would also prevent accurate satellite measurements of chlorophyll, it is more appropriate here to use the higher estimate of total cloud cover. Therefore, our results focus on the ISCCP cloud configuration. The ISCCP-simulated chlorophyll sampling strategy would be most comparable to a global network of geostationary satellites with passive ocean color instruments. Since this does not exist in the real world, we use a merged chlorophyll product that combines several polar-orbiting sensors to increase daily data coverage: the Ocean-Colour Climate Change Initiative (OC-CCI)

dataset, version 6.0 (Sathyendranath et al., 2019). This product combines chlorophyll data from SeaWiFS (Sea-viewing Wide-

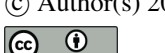



Field-of-view Sensor), MERIS (MEdium spectral Resolution Imaging Spectrometer), MODIS (Moderate-resolution Imaging Spectroradiometer, aboard the Aqua satellite), and VIIRS (Visible and Infrared Imaging Radiometer Suite). The data are available daily at 4 km spatial resolution.

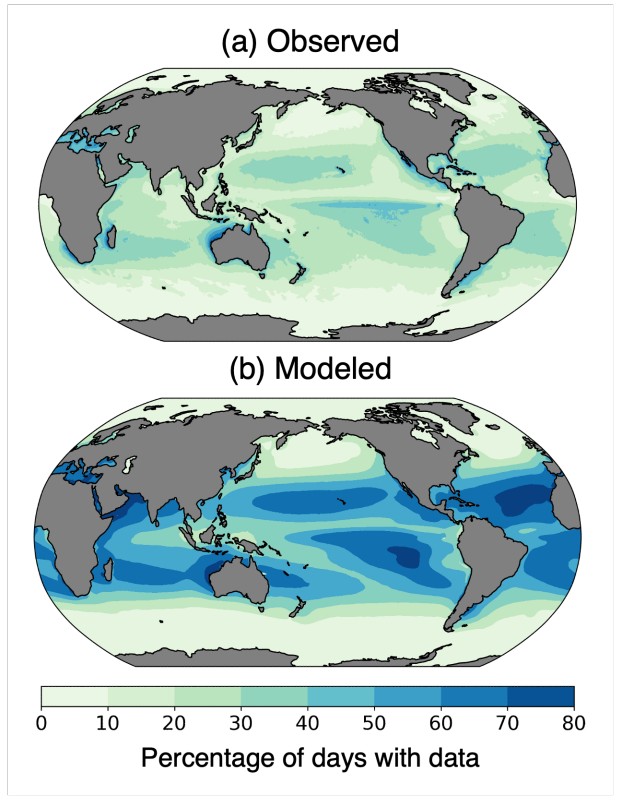

**Figure 4.** Percent of days with chlorophyll data coverage from (a) the Ocean Colour Climate Change Initiative (OC-CCI), 2006–2016, and (b) the pre-industrial simulation with the ISCCP configuration of ChlOSP.

As expected, the real world has more missing data than the simulated observations (Fig. 4). The OC-CCI chlorophyll product

has a median global daily coverage of 21 %, whereas the median daily coverage for the ChlOSP ISCCP configuration is 40 %. The ISCCP configuration of ChlOSP samples more frequently than real-world sensors, and it does not account for all conditions that prevent chlorophyll detection. Many of these factors are difficult to predict and model; for example, observations may be excluded from Level 3 data due to atmospheric correction failure, saturated observed radiance, stray light contamination, algorithm failures, or satellite navigation failure (Scott and Werdell, 2019). However, there are several factors that are

candidates for future versions of the simulator. Some atmospheric and oceanic constituents that prevent chlorophyll retrieval, such as white caps, coccolithophores, and aerosols are already simulated in some capacity in CESM and could be added to ChlOSP with minor modifications. Instrument-related challenges, such as sun glint and high sensor zenith angle, would also be valuable additions to future versions of ChlOSP. Since we are not accounting for all of these factors currently, we expect that



the total percent missing data will be lower in ChlOSP than in the real world. Therefore, our results represent a conservative
estimate of biases due to missing data on a global scale.

In addition to the viewing conditions built into the simulator, differences in missing data arise due to the modeled representation of Earth. Comparing the temporal coverage across the globe provides more insight into the distribution of missing data (Fig. 4). ChlOSP captures missing data in the subpolar and polar regions well. However, ChlOSP collects more data in the tropics and subtropics compared to the real-world observations. This is due to cloud biases in the model, as shown in Fig. 3.
We focus our study on the highly productive and very cloudy subpolar north Atlantic, Pacific, and Southern Ocean regions, where the modeled cloud bias is minimal; as such, ChlOSP is an appropriate modeling tool for our purposes.

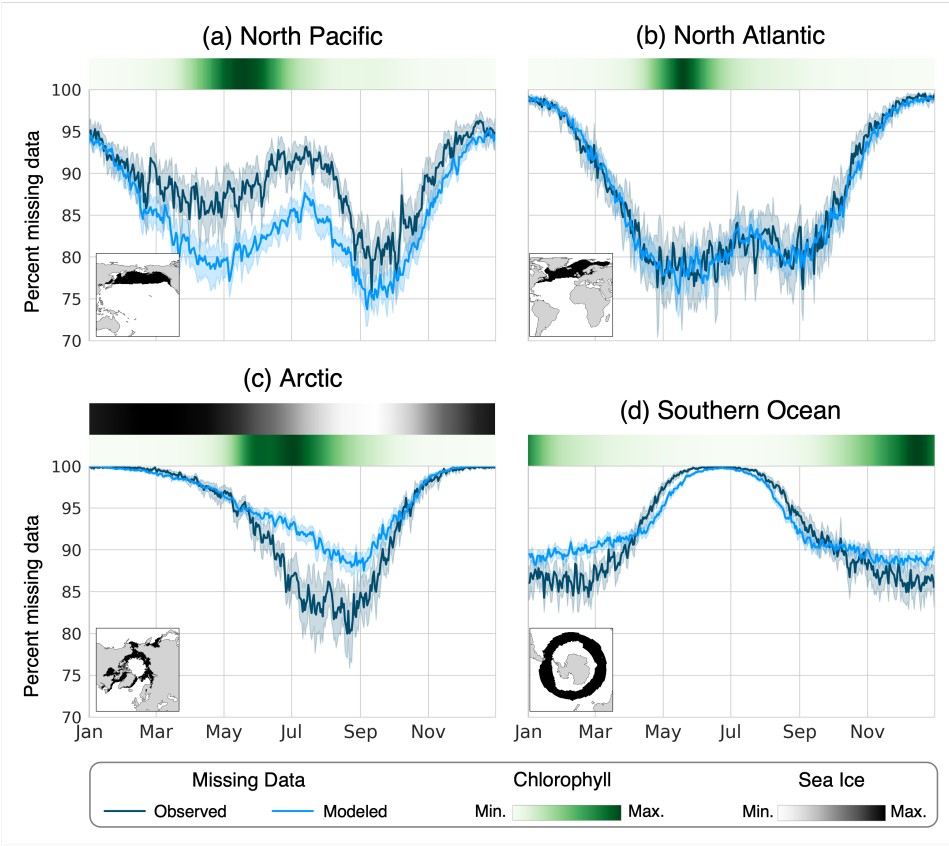

**Figure 5.** The mean percentage of area missing from the chlorophyll data for each day of the year in the (a) North Pacific, (b) North Atlantic, (c) Arctic, and (d) Southern Ocean. The observed (dark blue line) was calculated using the OC-CCI data product from 2006 to 2016, and the modeled (light blue line) was calculated from the 30 years of the pre-industrial simulation using the ISCCP configuration of ChlOSP. The error bars show the 95 % confidence intervals on on the daily mean. The green heatmaps show the seasonal cycle of chlorophyll derived from the Standard model output, and the gray heat map above the Arctic panel represents the sea ice seasonality.





Since we are interested in the seasonal cycle of chlorophyll in the productive and cloudy subpolar regions, accurately representing the seasonality of missing data is also important. The timing of missing data relative to the seasonal cycle in chlorophyll effects how the climatology is weighted in time. Figure 5 shows the mean percent area missing in four different

subpolar biomes for each day of the year. To provide seasonal context for each biome, the mean seasonal cycle of modeled chlorophyll is plotted as a heatmap above each panel, along with the mean sea ice fraction in the Arctic. For the model output, the weights were used to calculate the total area that was observed during the sunlit portion of the day. Overall, we find that the seasonality of missing data is appropriately captured in the model. Differences in the percent missing data between the model and real world may arise due to biases in modeled clouds and sea ice as well as the mean state of the climate. For example, the

modern-day satellite observations have lower sea ice coverage in the Arctic compared with the pre-industrial climate simulated in the model (Kay et al., 2022), leading to less missing data in the real world.

Another marked difference between real-world satellite observations and the full-field model output is the temporal variance in surface chlorophyll. There is more noise in the real world, due in part to missing data, which leads to lower autocorrelation in biome-averaged timeseries. Table 1 shows the e-folding timescale (number of days for the autocorrelation to reach 1/e) of

the de-seasonalized anomalies for satellite data compared to the Standard model output and the simulated observations from ChlOSP. While the ChlOSP output exhibits higher e-folding timescales compared with the satellite data, it shows improvement over the standard model output; the mean difference between e-folding timescales in our regions of interest is reduced from 86 to 3 days. Replicating autocorrelation and variance in a dataset is important for several statistical analyses, such as the time of emergence in a trend (discussed in the Applications section).

| Biome | Model: Standard | Model: ChlOSP | Satellite Data |
|---|---|---|---|
| Southern Ocean | 116.2 | 3.5 | 1.1 |
| North Pacific | 73.4 | 4.5 | 1.0 |
| North Atlantic | 60.8 | 4.1 | 0.8 |
| Arctic | 95.7 | 2.5 | 0.7 |

**Table 1.** e-folding timescales (in days) for chlorophyll timeseries data averaged over biomes of interest. The ChlOSP output represents the ISCCP configuration. The OC-CCI data product from 2006 to 2016 was used for the satellite data.

The amount of missing data in ChlOSP depends on both the simulator set-up (conditions used for masking) and the representation of physical variables in the model (concentration of clouds and sea ice). Despite some biases, our model evaluation demonstrates that, overall, ChlOSP realistically simulates the number of missing observations in a merged chlorophyll data product, particularly in regions with high biological productivity. We have also shown that the missing data leads to a more realistic modeled representation of the chlorophyll variance. Therefore, we can now use ChlOSP to investigate how this missing

data impacts our interpretation of chlorophyll climatology and seasonal cycles.



# 3 Results

## 3.1 Climatology and global mean

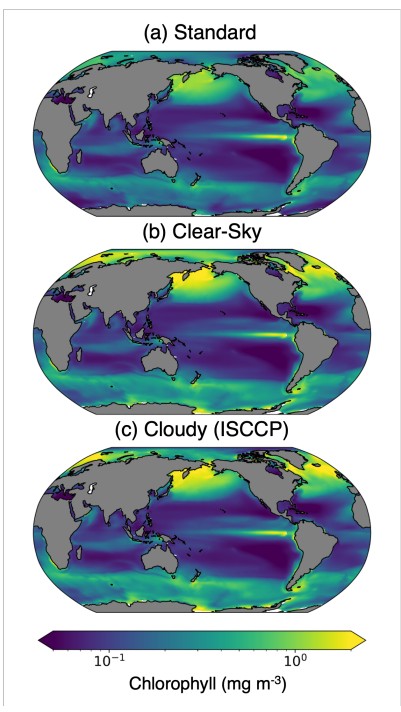

**Figure 6.** 30-year chlorophyll climatology from the pre-industrial simulation calculated with (a) Standard, (b) Clear-Sky, and (c) Cloudy (ISCCP configuration) model outputs.

A comparison of the Standard, Clear-Sky, and Cloudy (ISCCP) chlorophyll climatologies reveals that the temporal mean is impacted by missing data (Fig. 6). We highlight the differences between the various model outputs and configurations by calculating the percent differences in the climatology outputs (Fig. 7). To estimate the total sampling bias in simulated observations relative to the standard model output, we subtract the Standard climatology from the Cloudy climatology. We further estimate the contributions of sunlight and sea ice (Clear-Sky – Standard) and cloud cover (Cloudy – Clear-Sky) to the simulated observations of chlorophyll. The Cloudy minus Standard (Cloudy – Standard) maps indicate that sampling biases lead to >100 % overestimates of chlorophyll in the high latitudes. This pattern emerges largely from the Clear-Sky minus Standard maps, indicating that daylight-only sampling has the largest influence. Figure 7 reveals that the different configurations of ChlOSP show similar overall spatial patterns.

The greatest differences in the Clear-Sky minus Standard climatologies are located in the high latitudes, where there is insufficient light for satellite detection during the winter months. The winter months also correspond with low chlorophyll concentrations because the lack of sunlight limits phytoplankton growth. Additionally, in the polar regions, sea ice further

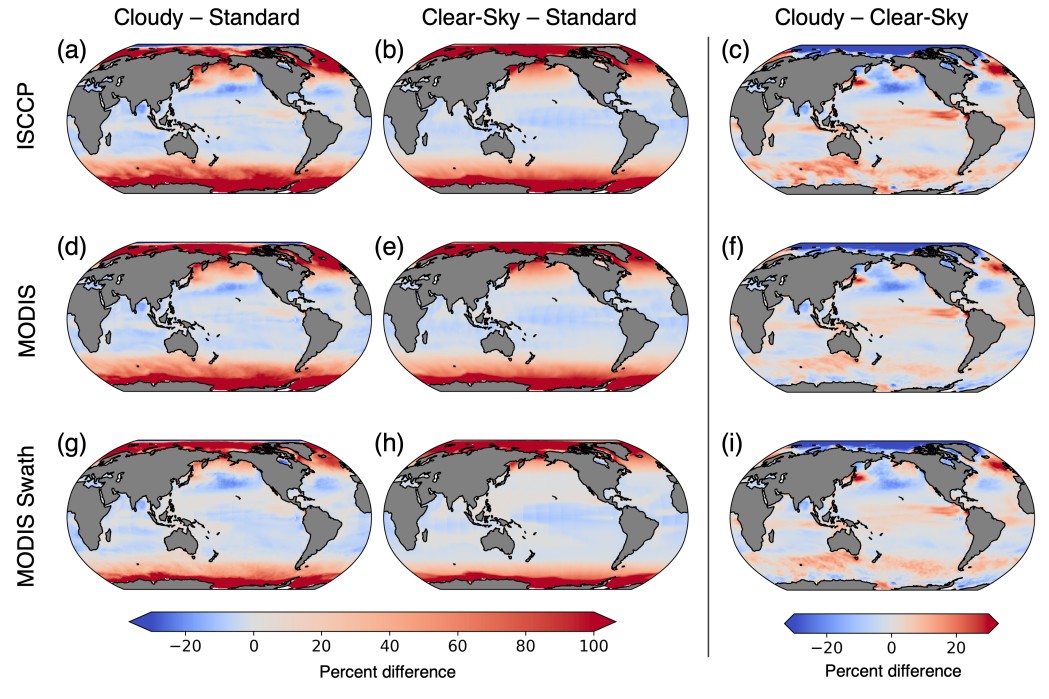

**Figure 7.** Percent difference in chlorophyll climatologies calculated with the 3 configurations of ChlOSP. The rows show the outputs from ISCCP, MODIS, and MODIS swath, respectively. The first column is the difference between the Cloudy chlorophyll and Standard. The second column is the difference between the Clear-Sky chlorophyll and Standard, which shows the impact of daylight-only sampling and sea ice on observations. The third column is the difference between the Cloudy chlorophyll and Clear-Sky chlorophyll, which isolates the impact of cloud cover on observations. Note that (b) and (e) are equivalent.

prevents satellite detection of chlorophyll during the start of the bloom season. As such, under-sampling in the winter leads to an overestimate of the mean chlorophyll concentration. In the low latitudes, a small underestimation of chlorophyll arises due to the phasing of the diurnal cycle relative to sampling time (Fig. 8). In Fig. 8, we select grid cells near the equator to illustrate how these sampling biases arise. This region exhibits the largest diurnal range, as shown in Fig. S5. Both the swath and daylight configurations of ChlOSP have a negative anomaly compared to the true mean (cf. yellow and gray dashed lines in Fig. 8).

A comparison of panels (e) and (h) in Fig. 7 reveals that implementing a satellite swath in ChlOSP impacts the Clear-Sky chlorophyll climatology. Overall, the biases are slightly less extreme when the swath is implemented: the differences decrease by about 9 % when averaged globally. In low latitudes, this is due to the sampling time. Fig. 8 reveals that, at the equator, the swath sampling has a smaller bias compared to the daylight sampling. In subpolar regions, the overestimate is also smaller when the swath is implemented (cf. Fig. 7 e and h). The swath version samples less frequently relative to the daylight configuration during the summer months (cf. Fig. S1 a and b), leading to weights that are more evenly distributed throughout the year.





Therefore, the bias towards the summer chlorophyll peak is less extreme in the Clear-Sky swath compared with the all-daylight version.

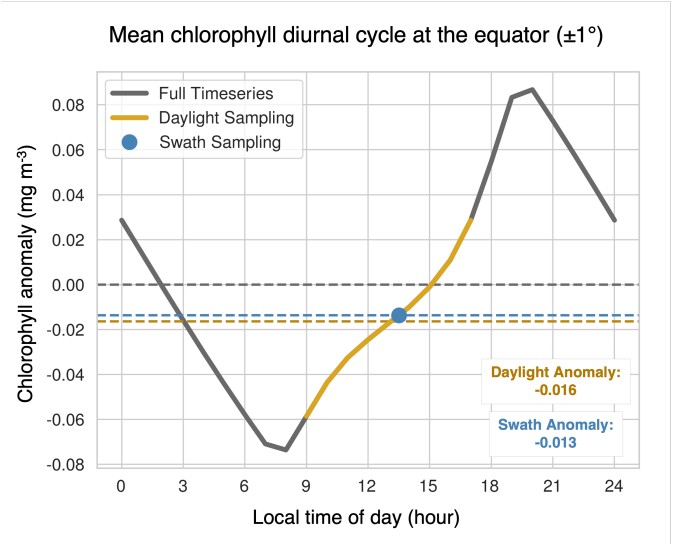

**Figure 8.** The mean diurnal cycle of chlorophyll at the equator over all months of the year, represented as chlorophyll anomalies. The gray line represents the entire cycle, the yellow line indicates the sunlit period where the solar zenith angle is less than 70 degrees, and the blue dot is the swath sampling time. The horizontal yellow and blue dashed lines correspond to the mean anomaly observed with daylight sampling and swath sampling, respectively, and the gray dashed line highlights the mean of the full period. To analyze the mean diurnal cycle, chlorophyll concentrations from the Standard output were grouped by local time of day (binned hourly) then averaged over all months and years. The time stamp of each data point is the end time of the averaging interval.

The Cloudy minus Clear-Sky (Cloudy – Clear-Sky) column in Fig. 7 isolates the impact of cloud cover on the simulated
observations of chlorophyll climatology. Clouds from the ISCCP simulator were used for panel (c) and clouds from the MODIS simulator were used for both panels (f) and (i). The impact of clouds is slightly sensitive to the sampling strategy (cf. panels (f) and (i)), but the total simulated cloud cover has a larger effect on chlorophyll climatology (cf. panels (c) and (f)). The difference in magnitude between panels (c) and (f) arises from the difference in total cloud coverage simulated by the two configurations (Fig. 3). Since the ISCCP simulator has more cloud coverage on average, there is a greater impact on the apparent chlorophyll
climatology. The spatial pattern in all three panels can be largely explained by the temporal correlation between cloud cover and Clear-Sky chlorophyll (Fig. 9). In regions where cloudy seasons correspond with lower chlorophyll (negative correlation), the chlorophyll climatology is overestimated relative to the Clear-Sky chlorophyll. Similarly, in regions where cloudy seasons correspond with higher chlorophyll (positive correlation), the chlorophyll climatology is underestimated. The impact of cloud cover is particularly important in the the Arctic, where clouds offset some of the large overestimates of chlorophyll due to
daylight-only sampling.



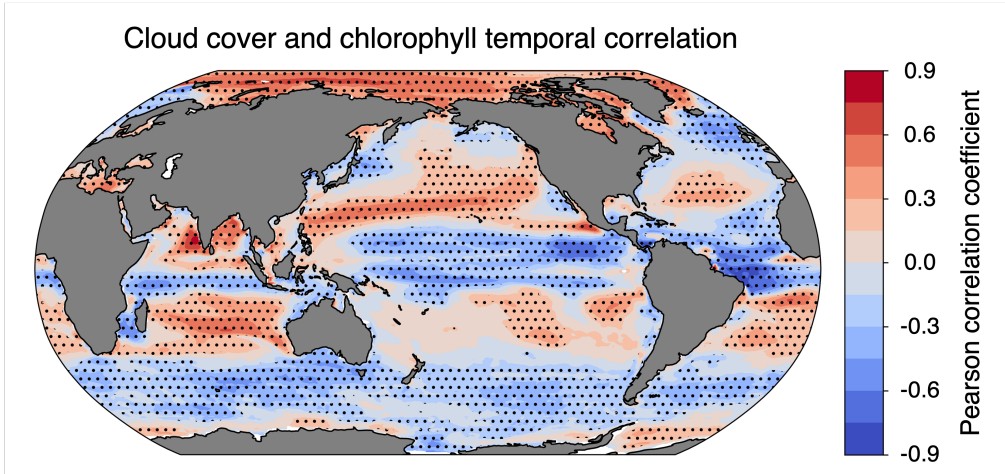

**Figure 9.** Pearson coefficient of correlation between monthly means of ISCCP cloud cover and Clear-Sky chlorophyll. Dotted cells indicate that the correlation was significantly different from zero at the 95 % confidence level. The effective sample size was determined by calculating the lag-1 autocorrelation for clouds and chlorophyll in each grid cell.

| Model Output | Global Mean Chlorophyll (mg m$^{-3}$) | Global Mean Difference from Standard | Standard Deviation (mg m$^{-3}$) |
|---|---|---|---|
| Standard | 0.215 | – | 0.553 |
| Clear-Sky | 0.245 | 14.1 % | 0.609 |
| Clear-Sky Swath | 0.263 | 22.2 % | 0.642 |
| Cloudy: ISCCP | 0.179 | -16.7 % | 0.463 |
| Cloudy: MODIS | 0.184 | -14.3 % | 0.477 |
| Cloudy: MODIS Swath | 0.200 | -7.4 % | 0.515 |

**Table 2.** Weighted global mean chlorophyll concentrations and weighted standard deviation for various configurations of ChlOSP. The weights are the product of the grid cell area and the ChlOSP weights (mean fraction of the grid cell that was "seen" by the satellite simulator). Note that chlorophyll is approximately log-normally distributed.

Global mean chlorophyll concentration estimated by ChlOSP is ∼20 % different from that estimated by the Standard configuration (Table 2, from Equation B3). From the maps in Fig. 7, the simulated observations appear to strongly overestimate global chlorophyll. However, this does not account for how often each location is sampled. The regions that show some of the highest sampling biases are sampled very infrequently due to cloud cover or lack of sunlight. If satellite sensors could see through clouds (as in the Clear-Sky configuration), the global chlorophyll mean would be overestimated by 14 to 22 %. However, including cloud coverage leads to an underestimation of chlorophyll, which ranges from -7 to -17 %. This is because locations with high chlorophyll values tend to also be cloudy, and therefore, they are sampled less frequently than other regions. The sampling frequency of the swath vs. daylight-only configurations also plays an important role. The Cloudy MODIS swath





configuration has a lower global bias than the daylight-only version (-7 % vs. -14 %). While the MODIS swath configuration
samples less frequently overall, it samples the high latitudes more frequently relative to other part of the globe (Fig. S3). This
is especially true in the summer months in high latitudes since the orbit passes over the poles many times per day. There-
fore, the biologically productive subpolar regions are weighted more strongly relative to other locations, leading to a smaller
underestimation in global chlorophyll.

### 3.2 Seasonal cycles

In addition to impacting the chlorophyll climatology, satellite-like sampling also impacts the spatially averaged chlorophyll
concentration. To investigate spatial means, we calculate the daily area-weighted chlorophyll mean within our biomes of
interest using the Standard, Clear-Sky, and Cloudy (ISCCP) outputs. The mean seasonal cycle was then calculated over the
30-year simulation (Fig. 10).

The largest differences between the Clear-Sky and Standard seasonal cycles occur during winter months, when ChlOSP
can not detect higher-latitude grid cells due to low light. These regions also correspond to low wintertime chlorophyll values
because there is limited light for photosynthesis. This difference between Clear-Sky and Standard is more pronounced in
biomes that span a larger latitude range, such as the North Atlantic. In the Arctic, sea ice also plays a major role in the apparent
seasonal cycle of the Clear-Sky chlorophyll. During most of the year, the Standard chlorophyll is lower than the Clear-Sky
in this region. However, in July and August, the Standard chlorophyll is higher than the Clear-Sky, indicating that there are
phytoplankton blooms beneath the sea ice that cannot be seen by ChlOSP.

Cloud cover also influences the apparent magnitude and, in some cases, the timing of the seasonal phytoplankton bloom.
The differences between Cloudy and Clear-Sky chlorophyll arise due to the spatial correlation between clouds and chloro-
phyll within each biome, which varies throughout the year. These correlations are shown in the boxes beneath each timeseries
(Fig. 10; the corresponding correlations for all biomes can be found in Fig. S6). Many biomes exhibit a positive spatial corre-
lation between cloud cover and chlorophyll concentration during the bloom months (e.g., Fig. 10a, b, c). Within these biomes,
model grid cells with high cloud cover tend to have higher chlorophyll, leading to lower biome-mean Cloudy chlorophyll con-
centrations relative to the Clear-Sky configuration (Fig. 10a, b, c). Conversely, in the Southern Ocean, there is a negative spatial
correlation between cloud cover and chlorophyll concentration through most of the year, leading to higher Cloudy chlorophyll
concentrations than the Clear-Sky configuration (Fig. 10d).

### 350 4 Applications

Our analysis so far has focused on using ChlOSP to assess how clouds, daylight vs. swath sampling, and the presence of sea
ice may bias satellite observations of chlorophyll. However, there are many other potential applications that we envision for
this tool.

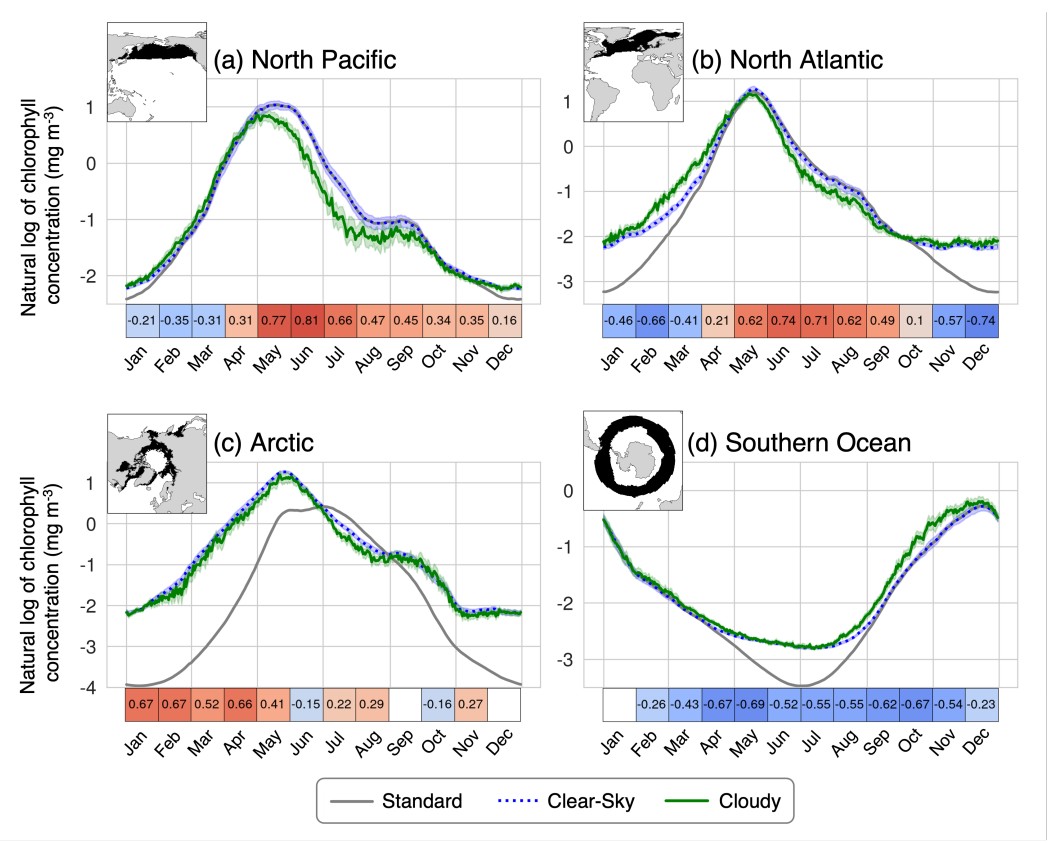

**Figure 10.** Seasonal cycle of Standard, Clear-Sky, and Cloudy chlorophyll (from the ISCCP configuration) in the (a) North Pacific (biome 2), (b) North Atlantic (biome 9), (c) Arctic (biomes 1 & 8), and (d) Southern Ocean (biome 16). The error bars on the Clear-Sky and Cloudy lines indicate the 95 % confidence interval on the mean. The boxes below the time series indicate the spatial correlation (Pearson's coefficient) between the mean cloud coverage and Clear-Sky chlorophyll within the biome for each month. White cells indicate that the correlation is not significantly different from zero at the 95 % confidence level. The effective sample size was calculated using Moran's I spatial correlation index.



## 4.1 Model tuning

Previously, tuning of the biogeochemical component of CESM has been accomplished by comparing Standard chlorophyll to real-world satellite observations. The goal is often to replicate the spatial pattern of the global chlorophyll climatology. However, as we have demonstrated, satellite observations of chlorophyll are biased due to missing data, whereas the Standard model output is not. Figure 11 compares the real-world, observed chlorophyll climatology from Aqua MODIS to the modeled 30-year pre-industrial climatology. Calculating the model bias using Standard chlorophyll vs. Cloudy outputs leads to different

results (cf. Fig. 11). Given that they are both impacted similarly by missing data, the Cloudy model output is more suitable for comparing model output with the real-world observations. These results indicate that the actual bias of CESM in the subpolar regions may be greater than previously thought, demonstrating the importance of taking sampling bias into account during the tuning process.

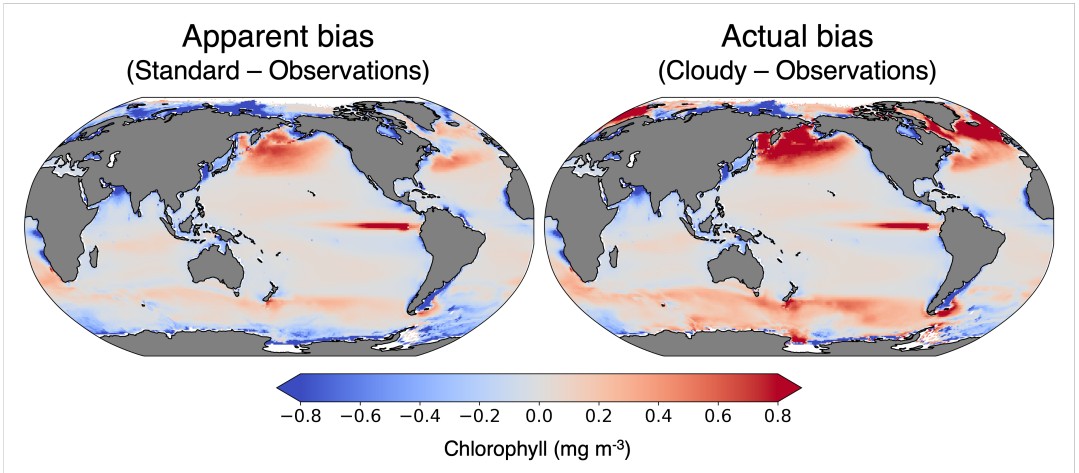

**Figure 11.** Model biases in chlorophyll climatology calculated with Standard and Cloudy (ISCCP) chlorophyll outputs. The observations are from Aqua MODIS from 2002 to 2023 and are re-gridded to the model resolution.

## 4.2 Net primary productivity

Another metric used for model tuning is the rate of globally integrated net primary productivity (NPP). NPP, the rate at which dissolved inorganic carbon is converted into organic matter, is particularly relevant for quantifying the global carbon cycle. The true modeled NPP can be determined directly by calculating the sum of the total carbon fixation vertical integral for all phytoplankton groups. This value is then compared to estimates of real-world marine NPP. There are many methods for estimating real-world NPP, many of which rely on satellite-observed chlorophyll. For example, the Vertically Generalized

Production Model (VGPM) uses chlorophyll along with sea surface temperature (SST) and photosynthetically active radiation (PAR) – also derived from satellite products (Behrenfeld and Falkowski, 1997).





We can make a more direct comparison between the model and the real world by calculating ChlOSP-estimated NPP (using VGPM) rather than the true model output (i.e., the vertically integrated phytoplankton carbon fixation). To calculate satellite-like NPP from ChlOSP, we use the chlorophyll, SST, and PAR climatologies in the VGPM algorithm. When integrating the

resulting NPP over the global oceans, we weight each grid cell area by the time-averaged satellite weights. We then calculate the total fraction of the ocean that was seen by the satellite and use this fraction to scale our NPP estimate to the full area of the ocean. The resulting NPP values are impacted by the version of chlorophyll in ChlOSP used as the input (Table 3). Since the Cloudy output is most similar to real-world satellite data, the 50.10 Pg C yr$^{-1}$ value should be used when tuning the model.

In addition to using simulated satellite-derived NPP to improve model tuning, we also demonstrate how ESMs can be used

as a testbed for NPP algorithms. In our CESM simulation, the true globally integrated NPP is 48.43 Pg C yr$^{-1}$. This value is remarkably similar to the VGPM-derived values, increasing our confidence in the accuracy of the real-world globally integrated NPP from VGPM.

| Chlorophyll Input used in VGPM | Global NPP (Pg C yr$^{-1}$) |
| --- | --- |
| Standard | 49.26 |
| Clear-Sky | 54.54 |
| Cloudy | 50.10 |

**Table 3.** Global net primary productivity calculated with VGPM model using ChlOSP (ISCCP) outputs.

### 4.3 Time of emergence

ChlOSP can also be used to calculate the time of emergence for chlorophyll trends in simulated observations. The time of

emergence is the length of the observational record required to identify a statistically significant trend within the context of internal variability. The impact of anthropogenic climate change on phytoplankton abundance is critically important to marine ecosystems and fisheries around the world. However, there is great uncertainty in global chlorophyll trends in the current satellite record (Beaulieu et al., 2013; Boyce et al., 2014; Gregg and Rousseaux, 2014; Hammond et al., 2017), due in part to the limitations of satellite data: the shortness of the record and the prevalence of missing data. Additionally, there is

further uncertainty in how surface chlorophyll trends translate to changes in total phytoplankton biomass (Siegel et al., 2013; Behrenfeld et al., 2016).

Using Earth systems models, we can project global phytoplankton biomass into the future using various forcing scenarios, where we know the true trend in surface chlorophyll and primary productivity. With ChlOSP enabled, we can also calculate the apparent trend from the simulated observations. Because we have a fully coupled model, we can account for any changes

in cloud cover due to warming. The time of emergence from the simulated observations gives us greater insight into when we might detect trends in the real world.

Since we have not yet generated a future projection with ChlOSP, here we compare the variability in monthly chlorophyll anomalies using the Cloudy and Standard outputs from our pre-industrial simulation. Throughout the majority of the globe,



the temporal variability is higher in the Cloudy output relative to the Standard (Fig. 12). Since the Cloudy dataset has more
noise, a longer timeseries is necessary to identify a statistically significant trend. To estimate the length of the timeseries
needed, we assumed a global trend in surface chlorophyll of $-5 \times 10^{-4}$ mg m$^{-3}$ per year and applied the method described in
Weatherhead et al. (1998). Our calculations (not shown) indicate that the time to emergence may be delayed by more than 10
years in the subpolar regions due to missing data. This preliminary analysis indicates that typical chlorophyll model outputs
may underestimate the time of emergence because they do not account for the enhanced variability in real-world observations.
Therefore, ChlOSP will be a valuable addition to future simulations by providing more realistic estimates of the time of
emergence.

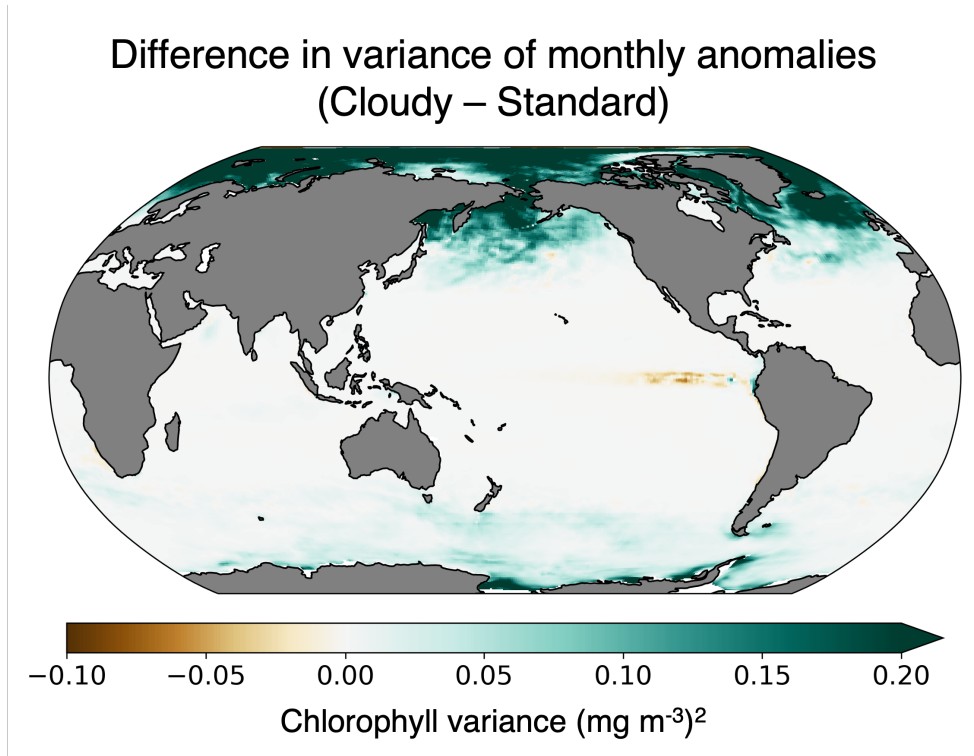

**Figure 12.** Difference in the variance of the monthly anomalies in surface chlorophyll concentration. The Cloudy output is from the ISCCP
ChlOSP configuration.

### 4.4 Gap-filling

Satellite-derived chlorophyll data are often gap-filled to generate a more complete dataset at high spatial and temporal resolu-
tion. A wide variety of methods have been applied to this problem: from simple linear interpolation to more complex methods
such as EOFs (Liu and Wang, 2018), neural networks (Krasnopolsky et al., 2016), and self-organizing maps (Jouini et al.,
2013). Stock et al. (2020) compared many of these methods within four study areas and found that ordinary kriging, spatiotem-





poral kriging, DINEOF, and random forests were the most successful methods, although results varied by region. Validating gap-filling methods is a challenge in the real world because we do not know what the truth is where we have missing data. This is often solved by transplanting artificial cloud masks onto clear-sky images or using data at a later time step. As we have shown, clouds and chlorophyll exhibit correlations, and there is a diurnal cycle in surface chlorophyll. Therefore, these method introduce additional error, making it difficult to quantify the error from the gap-filling method alone.

Here, we propose using an Earth system model testbed to apply various gap-filling techniques to the simulated observations from ChlOSP (Fig. 13). In the model world, we know the true chlorophyll values at every location, improving our ability to validate the gap-filled results. To generate gaps in the ChlOSP output, we use the weights as the probability of a grid cell being masked out. As an example, we have done a linear interpolation in Fig. 13. While the overall spatial pattern of chlorophyll is a close match between the gap-filled estimate and the model truth, smaller scale features are not captured well by the linear interpolation, particularly in regions with large amounts of missing data.

Future work will involve testing a wide variety of methods, with the goal of identifying the best method for gap-filling chlorophyll on a global scale. In the model, we have full knowledge of variables that impact phytoplankton growth – such as temperature, salinity, and wind – and can use this additional information in random forests or neural networks to predict chlorophyll. Many of these variables can be detected beneath cloud cover using microwave remote sensing (Gentemann et al., 2010), so these methods could be applied in the real world. One disadvantage of using the model as a testbed is that the resolution is much coarser than real-world satellite data, so it would likely not be suitable for gap-filling small-scale features. However, this method allows for the quantification of the error due to the gap-filling method.

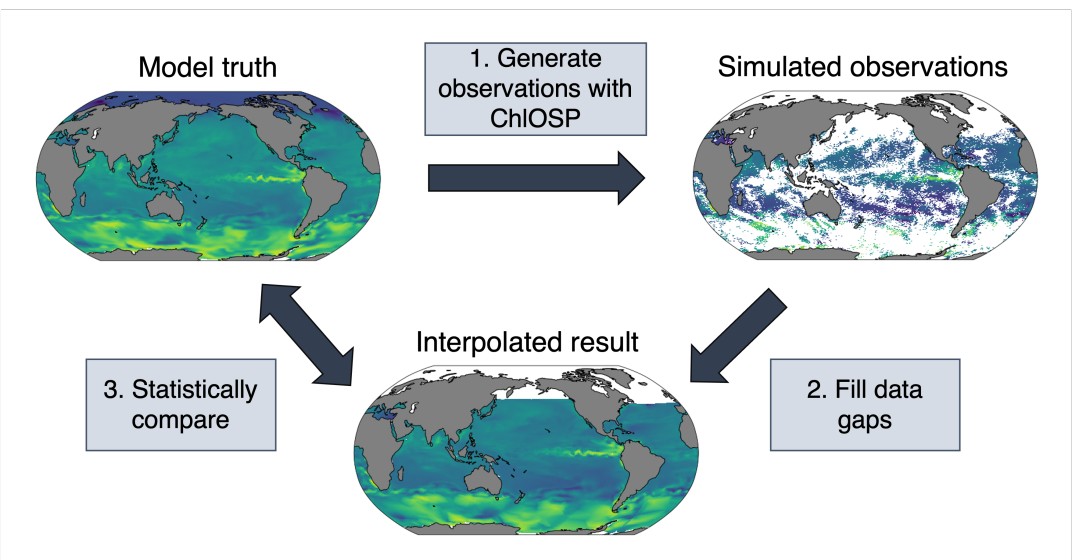

**Figure 13.** Diagram of the gap-filling testbed using ChlOSP-CESM.



## 5   Discussion and conclusions

We developed the Chlorophyll Observation Simulator Package (ChlOSP) for CESM, a fully coupled Earth system model. This new tool generates synthetic observations of surface ocean chlorophyll that are obscured by simulated cloud cover, sea ice, and high solar zenith angle. As a proof-of-concept, we ran ChlOSP in a 50-year pre-industrial simulation of CESM and analyzed the last 30 years. We tested three configurations of ChlOSP using different simulated cloud observations from COSP: ISCCP clouds, MODIS clouds, and MODIS clouds with an Aqua-like swath (1:30 pm sampling time). For each configuration, we compared the Cloudy (obscured by sea ice, high zenith angle and cloud cover), Clear-Sky (obscured by sea ice and high zenith angle only), and Standard (not obscured) chlorophyll outputs to assess the sampling bias that arises due to missing data. We found that missing data impacts the apparent climatology, overall global mean, and seasonal cycle in subpolar regions. We further demonstrated that ChlOSP can be used in future simulations to improve model tuning, calculate time of emergence of a trend, and test gap-filling methods.

We found that the simulated observations from the ISCCP configuration underestimate the true global mean chlorophyll by $\sim$0.03 mg m$^{-3}$, which is the same order of magnitude as the expected change by the end of the century ($\sim$0.05 mg m$^{-3}$; Schlunegger et al., 2020). The largest differences between the simulated observations and the Standard chlorophyll output are due to daylight-only sampling. These differences are particularly pronounced in the high latitude regions, where detection of low chlorophyll values during winter is not possible. This leads to an overestimation of mean chlorophyll over time, which is consistently over 100 % throughout the polar regions. In the real world, we have sparse in situ observations to compare with satellite data, therefore, our ability to estimate sampling bias is limited. However, our Fig. 7 results agree well with estimated real-world satellite sampling bias from Gregg and Casey (2007), where they applied satellite sampling to a global biogeochemical model with data-assimilated chlorophyll. Their results similarly showed that the largest sampling biases were due to the solar zenith angle threshold in high-latitude regions, with clouds being the second most important factor. They found positive biases in the annual global means (+8 %), which differs from the negative biases that we report (-16.7 %). However, this is because they report the standard mean rather than weighting by sampling frequency as we have done here.

Cloud cover plays an important role in the apparent mean of chlorophyll. Depending on the region, cloud cover can cause both positive and negative sampling bias. In many ocean regions, cloud cover and chlorophyll exhibit statistically significant correlations in both space and time (Fig. 9 and Fig. S6). Spatial correlations impact the mean chlorophyll over ocean biomes while temporal correlations impact the climatology in a given location. The mechanisms driving these correlations are not explored here, but we expect that this is a result of large-scale dynamics rather than a direct interaction between clouds and phytoplankton. In CESM, the PAR field at the ocean surface includes the influence of clouds, but the extent to which this impacts phytoplankton growth and/or photo-acclimation is not investigated here. There is some evidence that biogenic aerosols (DMS) produced by phytoplankton can increase cloud cover by acting as cloud condensation nuclei (Andreae and Crutzen, 1997), but this process is not represented in CESM. Further work is needed to investigate the direct effects of cloud cover on surface chlorophyll concentration and to validate the seasonal phasing of clouds and chlorophyll in the real world.





Through testing various configurations of ChlOSP, we found that the results are sensitive to both the definition of cloud cover and the sampling pattern. The ISCCP configuration had higher cloud cover than MODIS, which amplified the differences
between the Standard and Cloudy chlorophyll outputs (Fig. 7, Table 2). Since the ISCCP cloud simulator can detect partial cloud cover, it provides a more realistic representation of missing chlorophyll data. Therefore, we chose to focus the majority of our results on this configuration. Despite some differences in magnitude, we found that the overall patterns were consistent in all configurations.

Interestingly, compared with the all-daylight versions, the swath configuration led to greater chlorophyll biases in the Clear-
Sky global mean, yet smaller biases in the Cloudy global mean (Table 2). Fig. S1 highlights that in the swath configuration, the subpolar regions (generally more productive) are sampled more frequently than lower-latitude regions (generally less productive) – due to the polar orbit. Therefore, the swath configuration further enhances the overestimation of the global mean that arises in Clear-Sky chlorophyll due to summertime only sampling. However, when we add clouds, the swath sampling more accurately captures the global mean chlorophyll (Table 2). Fig. S3 reveals that when clouds are included, the resulting weights
are more evenly distributed throughout the globe because the productive subpolar regions tend to be cloudy. Additionally, we showed that at low latitudes, the 1:30 pm sampling time provides a more representative sample of the diurnal cycle than the all-daylight version (Fig. 8), which impacts the climatological mean of the Clear-Sky output (Fig. 7). This demonstrates the importance of simulating a realistic sampling pattern when assessing sampling biases in chlorophyll. Future work will involve implementing various swath widths, times, and orbital geometries to simulate a variety of sensors, including the upcoming
NASA PACE mission.

Our work focuses on sampling biases that arise due to missing data, but there are many other differences between modeled and observed chlorophyll. As shown in Dutkiewicz et al. (2018), large errors in observed chlorophyll – comparable in magnitude to what we found here – arise from the choice of algorithm used for estimating chlorophyll from remote sensing reflectance. We are unable to estimate these errors in CESM, as it currently lacks an optical model. While it is useful to isolate
the biases that arise due to certain factors, it would also be beneficial to understand the cumulative effect. We hope that ChlOSP will be implemented in an Earth system model capable of combining these various components.

While ChlOSP provides an improved model output for real-world comparison, it is not a perfect representation of satellite data. We have demonstrated that ChlOSP reasonably represents the amount of missing chlorophyll data (Fig. 4 and Fig. 5). However, ChlOSP does not include all factors that prevent satellite detection of ocean chlorophyll. Future improvements
to ChlOSP could involve adding more of these factors – such as white caps, coccolithophores, and aerosols – along with more realistic satellite orbits and associated sensor challenges, including sun glint and high sensor zenith angle. Additional discrepancies between missing data in the real world and ChlOSP arise due to the model's representation of the Earth system (i.e. cloud cover and sea ice in a pre-industrial vs. present-day climate).

The spatial resolution of the simulated observations from ChlOSP match the spatial resolution of the CESM configuration
used: in this case, ∼1 degree. This coarse spatial resolution is needed to run long-term climate simulations, but it is much lower than the resolution of satellite data. As such, ChlOSP does not capture small-scale heterogeneity that exists in the real-world, including coastal variability. Fig. 11 demonstrates how CESM does not resolve coastal regions, which tend to have high



chlorophyll concentrations. Therefore, this tool is best suited for large-scale, open-ocean analyses. To address partial cloud cover and sea ice within a model grid cell, we implemented a weighting method, which differs from the subcolumn strategy
utilized in COSP.

Currently, our ability to compare modeled and real-world chlorophyll is limited because ChlOSP was developed for a free-running, fully coupled climate model simulation. While this configuration permits future projections, internal variability complicates our ability to compare the model to the real world. We anticipate that the next version of ChlOSP will be implemented in a hindcast configuration, i.e. an ocean-only version of the model forced with momentum, heat, and freshwater fluxes from
historical observations spanning the duration of the satellite chlorophyll record. Using this version of the model along with in situ compliments, we plan to assess how clouds and missing data may have impacted our understanding of historical chlorophyll evolution.

Despite these uncertainties, our proof-of-concept simulation has demonstrated the utility of ChlOSP. The new model outputs allow for more robust comparisons between modeled and real-world chlorophyll, leading to improved model tuning and data
assimilation capabilities. Initial results indicate that there are differences in chlorophyll concentration between the typical model output and the satellite-like version. While we do not address all errors associated with ocean color remote sensing, we focus on one of the largest sources of uncertainty: missing data due to solar zenith angle and clouds. In the real world, we rarely know the true chlorophyll value where data are missing. However, in the model world, we know the exact values of all variables at every location and every time step, making it a powerful tool for estimating sampling bias. We anticipate that this
tool will open the door for a wide body of future work.

*Code and data availability.*  The model code is stored on github: https://github.com/genna-clow/CESM/tree/cesm2.2.0_satchl. Specific instructions for running CESM with ChlOSP can be found here: https://github.com/genna-clow/CESM/blob/cesm2.2.0_satchl/ChlOSP_guidance.md. The exact version of the model used to produce the results used in this paper is archived on Zenodo (DOI: 10.5281/zenodo.8071063). The data used to produce the figures are also archived on Zenodo (DOI: 10.5281/zenodo.8097543) and will be open access upon publication.

**Appendix A:  Model outputs**

We added the cloudfrac_modis and cloudfrac_isccp to the POP2 outputs for easier comparison to the other POP2 variables. These cloud fraction outputs include the daylight mask, which is written out as the cloudfrac_wgt variable. This output differs from the Clear-Sky weights because it does not include the weights from sea ice.

**Appendix B:  Equations**

For ChlOSP outputs, the climatology was calculated by applying the weighted mean. As an example, to calculate the mean of satellite-observed ISCCP chlorophyll over time, we use the model outputs totChl_isccp and totChl_isccp_wgt in Equation 1:





| | Name | Description |
|---|---|---|
| 1 | totChl | Sum of chlorophyll in the surface layer (10 m) over all phytoplankton functional types. |
| 2 | totChl_sat_nocld | Clear-Sky chlorophyll (daylight-only sampling and sea ice mask) |
| 3 | totChl_sat_nocld_wgt | Weights used for Clear-Sky chlorophyll |
| 4 | totChl_modis | Clear-Sky chlorophyll weighted by MODIS-simulated cloud cover. |
| 5 | totChl_modis_wgt | Weights used for totChl_modis |
| 6 | totChl_isccp | Clear-Sky chlorophyll weighted by ISCCP-simulated cloud cover. |
| 7 | totChl_isccp_wgt | Weights used for totChl_isccp |
| 8 | cloudfrac_modis | Total cloud fraction from MODIS simulator. |
| 9 | cloudfrac_isccp | Total cloud fraction from ISCCP simulator. |
| 10 | cloudfrac_wgt | Daylight weights for cloud fraction outputs. |
| 11 | totChl_sat_nocld_swath | Clear-Sky chlorophyll with 1:30 pm swath sampling |
| 12 | totChl_sat_nocld_wgt_swath | Weights used for Clear-Sky chlorophyll with 1:30 pm swath sampling |
| 13 | totChl_modis_swath | MODIS chlorophyll with 1:30 pm swath sampling |
| 14 | totChl_modis_wgt_swath | Weights used for MODIS chlorophyll with 1:30 pm swath sampling |
| 15 | cloudfrac_modis_swath | MODIS cloud fraction with 1:30 pm swath sampling |
| 16 | cloudfrac_modis_wgt_swath | Weights used for MODIS cloud fraction with 1:30 pm swath sampling |

**Table A1.** New model outputs from CESM-ChlOSP. These outputs are written out in a new file stream within POP2.

$$\text{weighted time mean} = \frac{\sum_t (totChl\_isccp(x,y,t))}{\sum_t (totChl\_isccp\_wgt(x,y,t))},$$ (B1)

where $totChl\_isccp = chlorophyll * weight$ (calculated within POP2) and $totChl\_isccp\_wgt = weight$. The chlorophyll and weight variables correspond to the surface chlorophyll concentration in a grid cell and the fraction of the grid cell observed during a given time step, respectively. Seasonal cycles were evaluated in ocean biomes (supplementary Fig. S4) using Equation 2:

$$\text{weighted spatial mean} = \frac{\sum_{x,y} (totChl\_isccp(x,y,t)) * TAREA(x,y)}{\sum_{x,y} (totChl\_isccp\_wgt(x,y,t) * TAREA(x,y))}.$$ (B2)

This is equivalent to taking the weighted average in space where the weights are equal to the total area observed (in square centimeters) within the biome at each time step. TAREA is the area of each model grid cell and is included in all POP2 output files. TAREA was subset for certain biomes of interest in these calculations. The total weighted mean over time and space is calculated with Equation 3:

$$\text{weighted total mean} = \frac{\sum_{x,y,t} (totChl\_isccp(x,y,t)) * TAREA(x,y)}{\sum_{x,y,t} (totChl\_isccp\_wgt(x,y,t) * TAREA(x,y))}.$$ (B3)



*Author contributions.* NL, KL, JK and ML conceptualized the study. ML and GC made code modifications and ran the model simulations. GC analyzed simulation results and prepared the manuscript. NL, KL, JK and ML assisted in preparing and reviewing the manuscript.

*Competing interests.* We declare no competing interests.

*Acknowledgements.* Computational resources were provided by the Computational and Information Systems Laboratory (CISL) at the National Center for Atmospheric Research (NCAR). We gratefully acknowledge the CESM Ocean Biogeochemistry Working Group allocating computational resources for this work. The simulation was run on NCAR's high-performance supercomputer, Cheyenne. This material is based upon work supported by the National Science Foundation Graduate Research Fellowship under Grant No. DGE 2040434, the University of Colorado Boulder Research and Innovation Seed Grant Program, and the University of Colorado Cooperative Institute for Research in Environmental Sciences Innovative Research Proposal Program. We would also like to acknowledge the COSP guidance provided through CFMIP: https://github.com/CFMIP/COSPv2.0/wiki/COSP2-in-CESM2-Guidance).





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
