# Peer review of "The utility of simulated ocean chlorophyll observations: a case study with the Chlorophyll Observation Simulator Package (version 1) in CESMv2.2"

_Geoscientific Model Development, 2023_

## Author Comment (AC1)

Thank you for taking the time to read our manuscript and provide constructive feedback. These comments were helpful for improving the paper. Please find our point-by-point responses to specific comments below. Reviewer comments are in black, followed by our responses in blue. Underlined portions indicate changes that were made in the manuscript. Line numbers refer to the revised manuscript.

**Response to RC2: 'Comment on gmd-2023-143', Anonymous Referee #2, 13 Oct 2023**

**Comment 1:** *My biggest issue is with my lack of understanding of the chlorophyll retrieval itself (how it is calculated and/or how the weights are used) and of its performance and validation.*
**Response:** This is valuable feedback, thank you. We have tried to clarify this throughout the manuscript —please refer to our responses to your comments below.

Importantly, we have modified the manuscript to provide additional clarity for the reader about how chlorophyll is 'retrieved' (i.e., calculated) in ChlOSP:

(Line 160) "To calculate the weight from modeled sea ice and cloud cover fields, which are both expressed in terms of the fraction of a grid cell that is covered, these values are subtracted from 1. All weights are assumed to be independent from one another, so the final weight is the product of the weights calculated from each input parameter. At every model time step, the surface chlorophyll is multiplied by the weights. Then, the weighted chlorophyll and the weights are both output by the model at the frequency specified by the user when running CESM (i.e. hourly, monthly, etc.). Both outputs are needed to calculate the weighted mean over space and/or time. The weighted chlorophyll is not a physical value that should be analyzed independently from the weights. When calculating the weighted mean of chlorophyll, the weighted chlorophyll output corresponds to the numerator and the weights output corresponds to the denominator in Equation 1:"

$$\text{weighted mean} = \frac{\sum_{i=1}^{n} w_i X_i}{\sum_{i=1}^{n} w_i}$$

To the reviewer's point about performance and validation, we focus our paper on a description of ChlOSP development, and we provide a validation of this development in the context of missing data. Figure 11 illustrates that the long-term mean modeled chlorophyll can differ quite a bit from that observed by satellite. We have a second study in the works with ChlOSP that will provide a detailed comparison of ChlOSP-generated chlorophyll and satellite chlorophyll across a wide range of spatial and temporal scales for the MODIS satellite period.

**Comment 2:** *I also left wondering about how the results would look if shorter time scales were considered (from daily to monthly to annually). For the most part, decadal averages are presented. This paints a great average picture, but shorter time scales are important too,*

*particularly regarding some of the applications of ChlOSP presented by the Authors.  It would be appreciated if they add a quick case study on shorter time scales.*

**Response:**  We appreciate the suggestion to look at shorter time scales. We have intentionally looked at longer time scales to account for the mismatch in the timing of interannual variability between the coupled model and the real world. When validating Earth system models, it is common to look at chlorophyll in terms of multi-year means when comparing to observations. We agree that shorter time scales would be important for applications, especially gap-filling, which will be addressed in a future paper. We have added a sentence in the abstract clarifying the goals of this paper (line 7): "Here, we introduce this new tool and present a preliminary study focusing on long timescales."

**Comment 3:** *Line 29: Since Hu uses a line-height approach, you might mention this in addition to just the blue-to-green ratio.*

**Response:** Thank you. We added a clarifying statement to this sentence (line 28): "The most commonly used algorithms to derive chlorophyll concentration from remote sensing reflectance rely primarily on the ratio of blue to green wavelengths."

**Comment 4:** *Line 139: The satellite "sees" a vertically optically weighted chlorophyll signal, not a sum of chlorophyll layers. Consider commenting on the difference between the remotely sensed signal and the sum that you use. Note also that in some waters, the satellite will not see 10 m deep in some (blue) wavelengths used in the retrieval algorithm.*

**Response:** Thank you for this comment. We have added a couple sentences addressing this in the introduction, where we discuss the differences between remotely sensed and modeled chlorophyll (line 55): "Satellite observations represent a vertically optically weighted chlorophyll signal, which is generally limited to the near-surface ocean due to light attenuation at depth. Therefore, the comparison with the vertically resolved model output is limited to the surface layer."

We also mention this assumption again in our description of ChlOSP (line 141): "We assume that the satellite can only see the surface layer of the POP2 grid, which represents depths from 0 to 10 meters. Although the depth seen by a satellite depends on the optical constituents present in the water column, the surface layer of the model roughly aligns with depths of in situ measurements used to validate the SeaWiFs chlorophyll retrievals (Gregg et al. 2004). The chlorophyll concentration in each surface model grid cell is calculated as the sum of chlorophyll from each phytoplankton functional group represented in MARBL."

**Comment 5:** *Lines 135 and 141: "missing" and "viewable" refers to present and absence of clouds, etc.?  Suggest clarifying in both places.*

**Response:** Thank you for this suggestion. We have added clarification that "viewable" means that the surface ocean is not obscured by clouds, sea ice, or low solar angles (line 147): "At each model time step, POP2 uses multiple variables to calculate the chlorophyll weights, which represent the fraction of each model grid cell that would be viewable by a satellite (i.e., the

fraction of the surface area that is not obscured)." In this way, "missing data" refers to those grid cells obscured by clouds, sea ice, and low solar angles.

**Comment 6:** *Line 144:  Suggest changing to "prevent satellite detection using passive instruments".*
**Response:** Agreed, we have updated this sentence: "ChlOSP accounts for clouds, sea ice, and low sunlight (high solar zenith angle), all of which prevent satellite detection using passive instruments."

**Comment 7:** *Line 150:  Change to "Level 3 ocean color retrievals have a high spatial resolution".*
**Response:** Changed to "Satellite-derived level 3 ocean color products have a high spatial resolution" since it may not be obvious that Level 3 refers to satellite data.

**Comment 8:** *Line 155:  I don't understand as written why you'd multiply the surface chl by the weights.  I get that you're deriving a fraction of viewable area in each 1-deg model cell that represents how much of that cell MODIS would see (0.5 indicating 50% of the 1-deg cell would be visible to MODIS).  Right?  But, applying that weight to the chl value would artificially lower the chl value, no?  As an example, if you're 1-deg cell has a chl value of 1.0 mg/m3 and that cell in reality is completely homogeneous, then even if half the cell was not viewable by MODIS, the satellite pixels would still have values of 1.0 mg/m3.  But, multiplying by 0.5 would lower the value to 0.5 mg/m3.  I have no doubt that I'm missing something, so please elaborate in this section.*
**Response:** We apologize for the confusion and are grateful that you brought this to our attention. We have clarified how the weights are calculated by including an equation along with several additional explanatory sentences in this paragraph:

"To calculate the weight from modeled sea ice and cloud cover fields, which are both expressed in terms of the fraction of a grid cell that is covered, these values are subtracted from 1. All weights are assumed to be independent from one another, so the final weight is the product of the weights calculated from each input parameter. At every model time step, the surface chlorophyll is multiplied by the weights. Then, the weighted chlorophyll and the weights are both output by the model at the frequency specified by the user when running CESM (i.e. hourly, monthly, etc.). Both outputs are needed to calculate the weighted mean over space and/or time. The weighted chlorophyll is not a physical value that should be analyzed independently from the weights. When calculating the weighted mean of chlorophyll, the weighted chlorophyll output corresponds to the numerator and the weights output corresponds to the denominator in Equation 1:"

$$\text{weighted mean} = \frac{\sum_{i=1}^{n} w_i X_i}{\sum_{i=1}^{n} w_i}$$

**Comment 9:** *Line 156: What does "desired frequency" mean?*
**Response:** Changed to "Then, the weighted chlorophyll and the weights are both output by the model at the frequency specified by the user when running CESM (i.e. hourly, monthly, etc.)."

**Comment 10:** *Line 203: Sorry if I missed it, but I think this is the first time you mention "time-average". Please elaborate.*
**Response:** We clarified the language here. The paragraph now reads: "When calculating the global mean of chlorophyll, we weight each grid cell by how frequently it was sampled (Equation B3). To do this, we calculate the time-mean of the weights, then multiply this by the area of each grid cell, which effectively represents the sample size for each grid point. Figure S3 shows the chlorophyll climatologies along with the corresponding time-mean of the weights for each cloudy configuration. The normalized weights represent the mean area seen by the satellite relative to other points on the globe."

**Comment 11:** *Figure 3: Suggested adding a 3rd row that presents a difference map.*
**Response:** Thank you for this suggestion. We have added a 3rd row to this figure.

[Figure]

**Comment 12:** *Line 230: Please identify from where these data were acquired.*
**Response:** We now include the link to OC-CCI data products, which is where we acquired the data (line 243): "...we use a merged chlorophyll product that combines several polar-orbiting sensors to increase daily data coverage: the Ocean-Colour Climate Change Initiative (OC-CCI, https://www.oceancolour.org/) dataset, version 6.0."

**Comment 13:** *Line 242:  Mentioning sun glint is important, thank you.  Consider elaborating on how omitting sun glint in this study might influence your results.  Given that sun glint is regional, I would expect meaningful impacts at low latitudes.  In practice for the future, you'd have to stratify your interpretation of results by sensor – SeaWiFS, e.g., tilts and saw far less sun glint than MODIS, which has complicated their intercomparison.*

**Response:** We added an additional sentence about sun glint and inter-orbit gaps impacting low latitudes. We cite the findings from Gregg & Casey 2007, which indicate that these factors do not play a significant role in chlorophyll biases in these regions. However, it could be interesting to test this with ChlOSP in a future study. As you mention, this would involve adding different model outputs for different sensors, so this may involve significant effort.

We have added this sentence at line 256: "Sun glint and inter-orbit gaps mainly impact low to mid-latitude regions. However, Gregg et al. 2004 found that chlorophyll sampling biases in these regions are small, so addressing these issues was not the primary goal of ChlOSP."

**Comment 14:** *Section 2.4:  I appreciate the careful evaluations presented, but I was surprised that direct-ish comparisons of satellite-derived and ChlOSP-derived chlorophyll were not presented. More specifically, I don't see much in the way of a performance assessment of the ChlOSP chlorophyll retrievals. Is there a reason for this?  Fig 11 touches on this, but in a highly averaged (20+ years) way that doesn't really validate the ChlOSP chlorophyll retrieval.  Could some short-term temporal subset of model output be resampled in a way using distributions of valid MODIS retrievals to create map(s) that enable a semi-direct comparison of chl in common bins? I realize that it won't be perfect (pre-industrial average vs. modern day MODIS), but at least seeing maps and their differences would provide the reader some knowledge of the scale of differences between the two chl retrievals.  Overall, I guess what I'm asking is, is there more than can be done to convince a reader that the ChlOSP chlorophyll retrievals are validated?*

**Response:** Thank you for this comment. We agree that comparison with real-world chlorophyll is important for CESM model validation — enabling this type of comparison was partly our motivation for developing ChlOSP. We focus our paper on a description of ChlOSP development, and we provide a validation of this development in the context of missing data. Indeed, Figure 11 illustrates that the long-term mean modeled chlorophyll can differ quite a bit from that observed by satellite. We have a second study in the works with ChlOSP that will provide a detailed comparison of ChlOSP-generated chlorophyll and satellite chlorophyll across a wide range of spatial and temporal scales for the MODIS satellite period.

**Comment 15:** *Line 278:  Not obvious to me how Figure 6 shows anything "temporal".  Would "long-term mean" be more appropriate wording?*

**Response:** We have changed "temporal" to "long-term" mean: "A comparison of the Standard, Clear-Sky, and Cloudy (ISCCP) chlorophyll climatologies reveals that the long-term mean is impacted by missing data (Fig. 6)"

**Comment 16:** *Table 2:  Might be informative to expand this table to include specific basins or regions?*

**Response:** Yes, thank you. We have added a supplemental table with this information. The table is referenced in line 324: "Global mean chlorophyll concentration estimated by the Cloudy ChlOSP output is ~20 % different from that estimated by the Standard configuration (Table 1; differences for individual biomes included in Table S1)."

| Biome | Standard | Clear-Sky | Clear-Sky Swath | Cloudy: ISCCP | Cloudy: MODIS | Cloudy: MODIS Swath |
|-------|----------|-----------|-----------------|---------------|---------------|---------------------|
| 1 | 0.511 | 205.2% | 139.0% | 208.3% | 204.5% | 147.1% |
| 2 | 0.770 | 40.3% | 18.2% | 16.7% | 19.8% | 5.7% |
| 3 | 0.349 | 11.4% | 1.7% | -13.0% | -10.1% | -14.7% |
| 4 | 0.100 | -2.3% | 0.4% | -8.6% | -8.1% | -6.0% |
| 5 | 0.097 | -3.2% | -1.6% | 0.1% | -0.6% | 0.5% |
| 6 | 0.356 | -5.1% | -5.5% | -0.1% | -2.8% | -2.7% |
| 7 | 0.089 | 0.4% | 1.5% | -9.6% | -8.8% | -7.6% |
| 8 | 0.357 | 141.9% | 125.4% | 99.1% | 107.7% | 90.0% |
| 9 | 0.644 | 73.7% | 53.5% | 68.1% | 69.6% | 54.6% |
| 10 | 0.454 | 17.5% | 2.4% | 0.9% | 2.1% | -5.8% |
| 11 | 0.109 | -2.9% | -1.0% | -4.9% | -5.7% | -4.0% |
| 12 | 0.145 | -6.5% | -6.5% | -0.4% | -1.7% | -2.5% |
| 13 | 0.141 | -1.8% | -1.9% | -5.8% | -6.4% | -6.4% |
| 14 | 0.112 | -2.7% | -2.2% | -5.6% | -5.6% | -4.9% |
| 15 | 0.342 | 23.2% | 3.4% | 29.3% | 27.3% | 11.5% |
| 16 | 0.220 | 59.1% | 32.0% | 92.7% | 77.4% | 49.4% |
| 17 | 0.148 | 162.2% | 134.8% | 244.7% | 206.4% | 179.7% |

**Table S1.** Mean chlorophyll values within ocean biomes (Figure S4), calculated using Equation B3. Column values are reported as percent difference relative to Standard.

**Comment 17:** *Line 318: Re: "the regions that show" – maybe a map of sample sizes would be useful?*
**Response:** We agree with this thought, but we think that this has already been sufficiently shown. Figure S3 shows the normalized weights, which conveys similar information. We have added a reference to that figure here: "The regions that show some of the highest sampling biases, such as subpolar biomes, are sampled very infrequently due to cloud cover or lack of sunlight (Fig. S3)." Additionally, Figure 4 shows the percentage of days with data, which indicates the sample size as well.

---

## Author Comment (AC2)

Thank you for taking the time to read our manuscript and provide constructive feedback. These comments were helpful for improving the paper. Please find our point-by-point responses to specific comments below. Reviewer comments are in black, followed by our responses in blue. Underlined portions indicate changes that were made in the manuscript. Line numbers refer to the revised manuscript.

**Response to RC1: 'Comment on gmd-2023-143', John Dunne, 20 Aug 2023**

**Comment 1:** *62 – what about the uncertainty associated with comparison of satellite optical depth and fully vertically resolved models?*
**Response:** This is an important point. We have added a sentence addressing this (line 55): "Satellite observations represent a vertically optically weighted chlorophyll signal, which is generally limited to the near-surface ocean due to light attenuation at depth. Therefore, the comparison with the vertically resolved model output is limited to surface layer only."

**Comment 2:** *Figure 1 – Cloud cover scale looks like it should only go down to 40%... how much of the oceans is the 20-40% range?*
**Response:** We agree that the map is clearer with the color scale bottoming out at 40%. We have updated this figure.

[Figure]

**Comment 3:** *70 – The mechanistic explanation is fairly simple and should be explained here – Areas where the ocean is cooler than the air (like upwelling regions) tend to cool the air, raise*

*the humidity, and form clouds. This is in addition to all ocean areas tending to raise the humidity.*

**Response:** We have added a sentence about this mechanism (line 73): "Ocean upwelling tends to cool the overlying atmosphere, which raises the humidity and leads to enhanced cloud cover. Upwelling also leads to increased nutrient concentrations, allowing more phytoplankton growth."

**Comment 4:** *187 – "equilibrium of the deep ocean can take hundreds of years" Actually, equilibrium of the deep ocean can take thousands of years.*

**Response:** Thank you for catching this. We have changed "hundreds" to "thousands".

**Comment 5:** *234 – "As expected"… It is not clear to me which of the model simulator versus real world differences make the strong difference in Fig. 4 "expected"… is it the return period of once a day in the model versus once every two days in the observations (line 178 "low latitude gaps are then filled during an orbit on the subsequent day"? More explicit attribution is needed here given the large number of differences described above… is the following sentence "The ISCCP configuration of ChlOSP samples more frequently than real-world sensors…" intended as the explanation? The connection to the previous discussion points is not clear.*

**Response:** Thank you for helping us clarify this paragraph. We have deleted "as expected". We also now point out that the mean daily coverage is higher for ChlOSP than in the real world, and we explain the reasons for this (line 248): "The ISCCP configuration of ChlOSP samples more frequently than real-world sensors because it samples at every sunlit time step rather than once-per-day." We then explain additional factors that result in less missing data, such as the lack of inter-orbit gaps and sun glint. The discrepancy between ChlOSP and real-world missing data is further explored in the subsequent paragraph where we discuss that an underestimation of cloud cover in the model (at low and mid latitudes) explains why ChlOSP samples more frequently at these locations compared with real-world satellites.

**Comment 6:** *241 – "white caps, coccolithophores, and aerosols are already simulated in some capacity in CESM and could be added to ChlOSP with minor modifications" if this would have improved the model with only "minor modifications" why was it not done in the present study?*

**Response:** We deleted "and could be added to ChlOSP with minor modifications." Although these factors are simulated in CESM, it would take significant effort to include these parameters in ChlOSP. Here we focus on the factors with the largest impact, and we believe that these contributions would be quite small compared with clouds, sea ice, and solar zenith angle.

**Comment 7:** *Table 1 – My interpretation of the longer e-folding time scales in the simulator than the observations is that the underlying model is missing important local scale forms of variance such as eddies, fronts, jets, etc. such that there is little correlation between the values separated two days apart. It is not clear the mechanism to which the authors are attributing this difference. In the simulator which is sampling daily, there is a resolved signal of autocorrelation. In the observations, there is not which contradicts the model behavior of signals persisting for 4 days. It is hard to know how to interpret the value of the model as an autocorrelation simulator in this case.*

**Response:** This is an excellent point. Thank you for bringing up your concerns. We previously had not considered how the spatial resolution may impact the e-folding time scales and hope to explore this in our next study. After further discussion, we decided that Table 1 was not necessary for validating the simulator and it has been removed from the manuscript.

**Comment 8:** *273 – "We have also shown that the missing data leads to a more realistic modeled representation of the chlorophyll variance." I do not see this assertion being supported anywhere as autocorrelation and variance are different things… has the variance in both the underlying model and simulator been assessed?*
**Response:** Thank you for mentioning this. Our language was inaccurate here, and we have deleted this sentence. This was referring to the e-folding timescale table that we have since deleted. In the Applications section, we assess the variance in monthly anomalies between the Standard and Cloudy chlorophyll outputs in Figure 12, in order to highlight the implications of variance in calculating the time of emergence.

**Comment 9:** *Figure 9 – I am not sure the meaning or value of this figure. The title refers to "chlorophyll temporal correlation, but I think the authors intend "chlorophyll bias correlation"… Are positive values where cloud cover leads to a positive bias in chlorophyll, and negative where it leads to a negative bias or is it truly the temporal correlation of high clouds when chlorophyll is temporally increasing as the title suggests? If the latter, than what is the significance? More clarity is needed.*
**Response:** We apologize for our lack of clarity here. We changed the title of the figure to highlight that we are taking the correlation of the seasonal cycles. We have also added a sentence to further explain the relevance of this figure (line 316): "The spatial pattern in all three panels can be largely explained by the correlation between the seasonal cycle of cloud cover and Clear-Sky chlorophyll (Fig. 9). The similarity in the spatial patterns between Figure 9 and Figure 7c suggest that correlations on monthly timescales play an important role in the resulting cloudy chlorophyll climatology."

[Figure]

**Comment 10:** *319-320 – "If satellite sensors could see through clouds (as in the Clear-Sky configuration), the global chlorophyll mean would be overestimated by 14 to 22 %." Is this due to the diurnal sampling bias, the seasonal sampling bias, or something else? The following*

*sentences discuss mechanisms and may be connected. Answering this may simply involve switching the order of the sentences such that the mechanisms are attributed directly to the result… e.g. "… As a result, if satellite sensors could see through clouds (as in the Clear-Sky configuration), the global chlorophyll mean would be overestimated by 14 to 22 %."*

**Response:** Thank you for helping us make this paragraph more clear. We have added a sentence to clarify why the Clear-Sky mean is higher (line 328): "This is because the Clear-Sky mean is heavily biased towards the productive summer months in high latitude regions due to solar zenith angle limits in the wintertime."

---

## Author Response (AR2)

gmd-2023-143
November 30th 2023

**Author's Response**

We would like to thank the editor and both reviewers for their time. We appreciate the feedback that we have received. The manuscript has been edited as indicated in our responses to reviewer comments. Additionally, we have made minor wording changes throughout the manuscript to clarify certain points.